# PROMEDIATE: A SOCIO-COGNITIVE FRAMEWORK FOR EVALUATING PROACTIVE AGENTS IN MULTI-PARTY NEGOTIATION

## ABSTRACT

While Large Language Models (LLMs) are increasingly used in agentic frameworks to assist individual users, there is a growing need for agents that can proactively manage complex, multi-party collaboration. Systematic evaluation methods for such proactive agents remain scarce, limiting progress in developing AI that can effectively support multiple people together. Negotiation offers a demanding testbed for this challenge, requiring socio-cognitive intelligence to navigate conflicting interests between multiple participants and multiple topics and build consensus. Here, we present PROMEDIATE[1], the first framework for evaluating proactive AI mediator agents in complex, multi-topic, multi-party negotiations. PROMEDIATE consists of two core components: (i) a simulation testbed based on realistic negotiation cases and theory-driven difficulty levels (PROMEDIATE-Easy, PROMEDIATE-Medium, and PROMEDIATE-Hard), with a plug-and-play proactive AI mediator grounded in socio-cognitive mediation theories, capable of flexibly deciding when and how to intervene; and (ii) a socio-cognitive evaluation framework with a new suite of metrics to measure consensus changes, intervention latency, mediator effectiveness, and intelligence. Together, these components establish a systematic framework for assessing the socio-cognitive intelligence of proactive AI agents in multi-party settings. Our results show that a socially intelligent mediator agent outperforms a generic baseline, via faster, better-targeted interventions. In the PROMEDIATE-Hard setting, our social mediator increases consensus change by 3.6 percentage points compared to the generic baseline (10.65% vs 7.01%) while being 77% faster in response (15.98s vs. 3.71s). In conclusion, PROMEDIATE provides a rigorous, theory-grounded testbed to advance the development of proactive, socially intelligent agents.

## 1 INTRODUCTION

Large Language Models (LLMs) are now widely integrated into agentic frameworks to assist individual users in completing diverse tasks such as information seeking and social skill development (Yang et al., 2024a; Shaikh et al., 2024; Eigner & Händler, 2024). Although these agent applications designed for individual users have shown promise, they contrast with real-world scenarios in which group collaboration among multiple users is necessary to drive results (Marks et al., 2001; Kozlowski & Ilgen, 2006; Li et al., 2024). This gap highlights a growing need for agents capable of proactively managing multi-party interactions and facilitating collaborative workflows. Prior research on AI agents in multi-party scenarios has either focused on qualitative analyses of proactive agents (Houde et al., 2025; Alsobay et al., 2025) or on reactive agents that provide assistance only when explicitly prompted (Chiang et al., 2024; Chiang, 2025). While some proactive agents have been designed for multi-party settings (Houde et al., 2025; Wesche & Sonderegger, 2019), systematic evaluation methods to guide and measure progress in this domain remain scarce. Developing evaluation frameworks is a critical and timely challenge for advancing proactive multi-party AI.

Multi-party conversations are inherently complex and demand more than the ability to solve a task: they require *socio-cognitive intelligence* to track multiple perspectives, anticipate divergent partici-

---

[1]The code will be released upon publication.

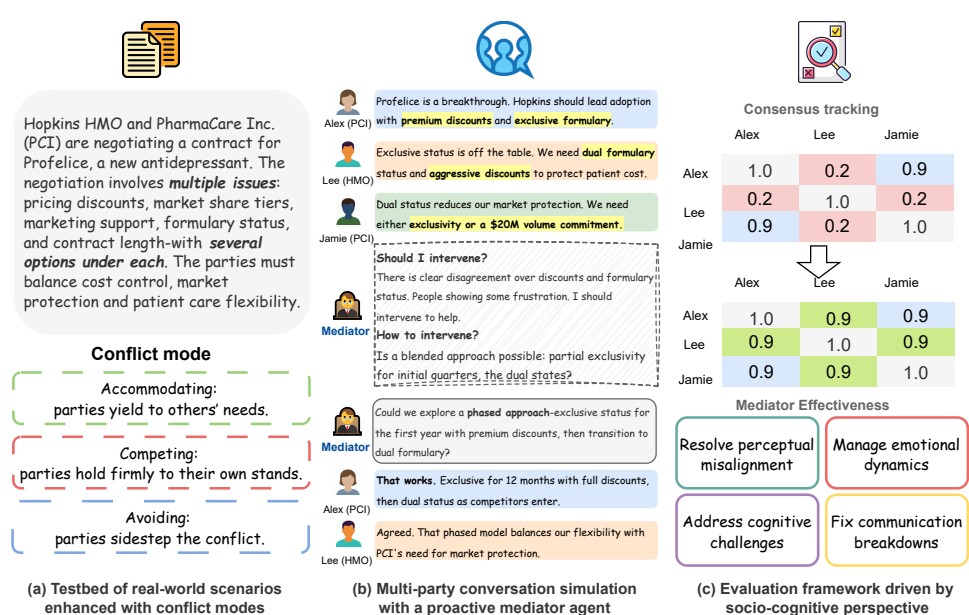

Figure 1: The illustration of **ProMediate** framework, involving a **multi-topic, multi-option** negotiation scenario with different conflict modes (Cai & Fink, 2002); conversation simulation with a plug-and-play agent; a suite of socio-cognitive evaluation metrics to capture the evolving nature of the negotiation.

pant attitudes, and *proactively* steer the discussion toward shared outcomes. Consider a *negotiation scenario* where a pharmaceutical company (PCI) and a health maintenance organization (HMO) are neogiating a contract for a new drug but have reached an deadlock, showing very low consensus (Figure 1). In such a scenario, a proactive AI agent must intervene promptly when perceptual, cognitive, emotional, or communication breakdowns happen and effectively guide the participants away from deadlock and toward a mutual consensus across various contract topics, like pricing discounts and formulary status. Existing AI benchmarks typically rely on simplified, game-based settings Abdelnabi et al. (2024); Bianchi et al. (2024) and evaluate social intelligence in bilateral (one-on-one) interactions Zhou et al. (2023), which overlooks such complex socio-cognitive dynamics of multi-party interactions.

To address these gaps, we introduce ProMediate (Figure 1), the first evaluation framework designed to evaluate proactive agents in complex multi-party conversations. ProMediate consists of two components: (1) a testbed for simulating realistic multi-party negotiations grounded in real-world cases and theory-based conflict modes, featuring simulated humans with structured preferences and a plug-and-play proactive AI mediator; and (2) a socio-cognitive framework for systematically measuring the negotiation success, AI mediation effectiveness and intelligence. A key feature of our simulation is that the AI mediator act proactively – the AI mediator must decide both *when to intervene* and *how to intervene*.

To evaluate socio-cognitive intelligence, we introduce a suite of metrics that measure consensus change, topic-level efficiency, AI response latency, mediator effectiveness in affecting the consensus, and mediator intelligence across perceptual, emotional, cognitive, and communicative dimensions. We evaluate three settings – NoAgent (negotiaion with no mediator agent), Generic Mediator, and Socially Intelligent Mediator – across six scenarios and three difficulty levels (ProMediate *Easy*, ProMediate *Medium*, and ProMediate *Hard*). In the Hard setting (with *Competing* participants), the Socially Intelligent Mediator delivers larger consensus gains than the Generic Mediator (+3.6%) while responding about 3× faster. Scenario difficulty strongly modulates outcomes: Easy setting (e.g., with *Accommodating* participants) yield larger, steadier gains. In contrast, Hard conflict modes are more volatile but benefit the most from proactive AI mediation compared to NoAgent. Finally, our evaluation metrics align with human judgments and reveal two key dimensions of success: consensus/efficiency and intervention tempo, supporting construct validity.

Ultimately, our work highlights that no single metric can capture a mediator's full capabilities, demonstrating the need for a multi-faceted evaluation approach for proactive agents in real-world multi-party scenarios. Our contribution are as follows:

- **Extensible testbed:** We design a challenging benchmark that captures the complexity of real-world multi-party interactions while providing a plug-and-play framework that allows the community to seamlessly integrate and evaluate different agents within a unified environment.
- **Integration of socio-cognitive framework:** We ground both our agent design and evaluation metrics in socio-cognitive theory, enabling a principled assessment of mediation skills and socially intelligent behavior.
- **Comprehensive evaluation and analysis:** We propose systematic metrics that illuminate agent capabilities from multiple dimensions, offering clear insights into strengths and limitations.

## 2 PROMEDIATE TESTBED

To evaluate proactive mediation in multi-party negotiations, we design a testbed that combines realistic simulations with configurable mediator interventions. We introduce the negotiation scenario setup, followed by multi-party conversation simulation with plug-and-play agents.

### 2.1 NEGOTIATION SCENARIO SETUP

To ensure conversational complexity and diversity, we adopt negotiation training materials from Harvard Law School's Program on Negotiation (pon.harvard.edu/store). These materials encompass negotiation-related scenarios spanning diverse domains and provide comprehensive instructions that typically require 2-3 hours for students to complete. We selected six scenarios covering multiple topics in healthcare, environmental policy, business development, etc., **with each scenario featuring multiple parties, multiple topics and multiple options for each topics.**

We formally structure each scenario as a multi-party negotiation framework with $N$ parties $\{P_1, P_2, \ldots, P_N\}$ discussing $M$ distinct topics $\{T_1, T_2, \ldots, T_M\}$. For each topic $T_j$, there exists a finite set of $S_j$ available options $O_j = \{o_{j,1}, o_{j,2}, \ldots, o_{j,S_j}\}$. Each party $P_i$ maintains an explicit preference ranking in the beginning. For instance, party $P_1$'s preference ordering for topic $T_1$ will be represented as $o_{1,2} > o_{1,3} > o_{1,1}$, indicating that option $o_{1,2}$ is most preferred. During conversation initialization, all background knowledge and individual preference profiles are incorporated into each agent's memory system. Table 4 in Appendix B.1 shows a sample scenario.

### 2.2 CONVERSATION SIMULATION

For the human simulation component, we build upon the InnerThought framework (Liu et al., 2025a), which enables LLMs to proactively participate in conversations by generating internal thoughts and using intrinsic motivation scores to select the next speaker (more details about the framework is in Appendix B.2). The original formulation, however, is designed for chit-chat: it lacks (i) explicit agenda/topic state for multi-topic tasks, and (ii) a principled policy for when and how a mediator should intervene. In our setup, we provide each simulated human agent with a detailed negotiation context and a structured, pre-specified preference profile (Section 2.1). Agents also have explicit identity profiles and employ a negotiation-driven reasoning process for determining and when and how they should intervene.

**Plug-and-play Mediator agent**   We design a **plug-and-play mediator** agent by clearly defining *When to intervene* and *How to intervene*, as shown in Figure 1(b). The mediator continuously observes the conversation and determines whether intervention is needed. If an intervention is warranted, it generates a response, and other simulated humans are skipped for that turn. Otherwise, the next speaker is chosen through the InnerThought framework, selecting the human with the highest "motivated thought". This design keeps the mediator modular and independent, avoiding the complexity of joint orchestration with humans. Moreover, it isolates the mediator's contribution, making its effect on the multi-party collaboration easier to evaluate and measure as we demonstrate in Section 4. All the prompts are shown in Appendix E.

**Conflict Modes** Prior work shows that assigned personas/roles systematically shape how conversations unfold—affecting style coordination (Thomas, 2008; Zhang et al., 2018), engagement, and outcomes. Guided by this, we instantiate persona at the group level via shared conflict modes to enrich conversational diversity across scenarios. We incorporate multiple conflict modes inspired by existing theory [2] (Cai & Fink, 2002; Ma, 2007; Thomas, 2008) :

- **Competing**: parties adopt firm positions and prioritize their own interests.
- **Avoiding**: parties strategically sidestep contentious topics and resolve the easier ones first.
- **Accommodating**: parties are receptive to others' views and willing to cooperate when necessary.

In Section 4, we use these modes to create the PROMEDIATE-Easy, PROMEDIATE-Medium, and PROMEDIATE-Hard difficulty levels for our evaluation framework.

## 3 PROMEDIATE METRICS

We evaluate proactive mediators in simulated multi-party conversations through a socio-cognitive lens, assessing two key dimensions: (1) group consensus dynamics as a socio-cognitive outcome—tracking how the agreement emerges and fluctuates throughout the negotiation; and (2) the mediator's socio-cognitive intelligence —assessing mediation skills. We first detail the consensus-tracking algorithm, then describe the socio-cognitive concept used for intelligence evaluation.

### 3.1 CONSENSUS TRACKING

In mediation and multi-party negotiation, *consensus* is not merely a procedural end state but a socio-cognitive achievement that emerges as parties share attitudes, align interpretations, and reach agreement through interaction (Swaab et al., 2007; Levine, 2018; Butera et al., 2019). Negotiation is a collective process, so individual success rates are insufficient. Multi-topic talks rarely end in full agreement as different parties hold different stances on each topics, and defining consensus as a unanimous binary is overly restrictive (del Moral et al., 2018). Unlike prior research that focused solely on final performance outcomes (Fu et al., 2023; Abdelnabi et al., 2024), our work introduces *consensus tracking*—a soft, time-varying measure that captures how individual attitudes shift and how agreement among parties emerges and evolves throughout mediator-guided interactions.

Consensus tracking consists of two components as shown in Algorithm 1: *(i) attitude extraction* and *(ii) agreement scoring*. Attitude extraction can be approached in several ways, such as probing a speaker's latent mental states or estimating preferences over pairs of options (del Moral et al., 2018). However, real-world conversations pose practical challenges that are often overlooked: (i) the option set is open—new alternatives may be introduced mid-conversation—so methods assuming a fixed inventory are brittle; (ii) internal mental states can diverge from the attitudes perceived by others; and (iii) not every topic is mentioned at every turn, making turn-level "overall" attitude estimates ill-posed. To address these issues, we use an LLM (GPT-4.1) to infer, from utterance text only, each participant's stance on each topic at each turn, yielding topic-specific, turn-conditioned attitudes without relying on fixed option sets or unobservable mental states (Prompts are shown in Appendix E). At initialization, participants are provided with their preference for options for all topics, giving us a complete attitude profile. At each subsequent turn, we extract the participant's updated attitudes from their new utterances: if a topic is mentioned, the attitude is updated; otherwise, the previous attitude is retained.

For agreement scoring, we compute pairwise agreement scores based on extracted attitudes between parties on each topic and then average these scores across all pairs to obtain a group-level measure. We employ an *LLM-as-a-judge* approach, prompting GPT-4.1 to assign an agreement score in the range $[0, 1]$ along five dimensions we create according to multple socio-cognitive theories (Griffiths et al., 2021; Thomson et al., 2009; Bedwell et al., 2012):

- **Shared goals:** Do both parties express alignment on the overall objective?
- **Common Understanding**: Is there a shared understanding of the problem and its context?
- **Agreement on Terms**: Do both parties accept the proposed terms and converge toward a shared resolution?
- **Tone and willingness**: Is there an evidence of cooperative tone, openness to compromise?

---

[2]https://www.psychometrics.com/conflict-resolution-skills-how-to-get-the-best-of-each-of-the-five-modes/

---

**Algorithm 1** Consensus Tracking

---

**Require:** Initial attitudes $Attitude_0[P_i][T_m]$ for each participant $P_i$ and topic $T_m$
**Require:** Conversation turns $C = [(P_1, u_1), (P_2, u_2), \ldots, (P_N, u_N)]$
  1: Initialize agreement scores $A_0[P_i][P_j]$ for all participant pairs $(P_i, P_j)$
  2: **for** each turn $(P_i, u_i) \in C$ and each topic $T_m \in T$ **do**
  3:     **if** $T_m$ is mentioned in utterance $u_i$ **then**
  4:         Update $Attitude_i[P_i][T_m]$ based on attitude extraction of $u_i$
  5:     **else**
  6:         $Attitude_i[P_i][T_m] \leftarrow Attitude_{i-1}[P_i][T_m]$
  7:     **for** each participant $P_j \neq P_i$ **do**
  8:         Compute agreement score $A_i[P_i][P_j][T_m]$ as the average over five dimensions using LLM-as-a-judge

---

- **Shared decision making**: Do both parties share the similar decision making process?

We also evaluate alternatives for agreement scoring—representing agreement on a single dimension and incorporating temporal context by providing the previous turn's agreement score when predicting the current one. These methods yield similar trends in consensus dynamics, and we provide additional details in the Appendix C. In the experiments, the agreement scoring is based on multiple dimensions and current context only.

## 3.2 SOCIO-COGNITIVE INTELLIGENCE

Successfully mediating and resolving conflict impasses requires strong socio-cognitive intelligence. To measure this, we take inspiration from socio-cognitive frameworks to evaluate the intelligence of mediators. We operationalize problems which could happen in a negotiation along four dimensions, adapted from the mediation theory matrix (Zariski, 2010):

- **Perceptual Differences**: divergences in beliefs, interpretations, or framings of key issues.
- **Emotional Dynamics**: negative emotions (for example, anger, distrust, grief) that derail constructive engagement.
- **Cognitive Challenges**: reasoning failures or biases (e.g., anchoring, confirmation bias) and limited option generation.
- **Communication Breakdowns**: ineffective exchange, talking about one another, hostility / escalation or nonresponsiveness.

We evaluate whether the mediator can recognize those key problems in the negotiation and propose effective strategies to resolve them.

## 3.3 EVALUATION METRICS

Building on the consensus tracking and socio-cogntive intelligence, we report metrics from two complementary perspectives: (i) *conversation-level outcomes* that characterize consensus dynamics independently of any mediator, and (ii) *mediator-level effectiveness*, evaluating whether interventions are effective and yield measurable improvements in consensus:

- **Consensus Change (CC)** We measure this as the improvement in consensus from the start to the end of a dialogue, aggregated over all participants and topics. If the mediation is effective, we should see a large Consensus Change. Since consensus constantly fluctuates, to reduce noise and outliers, we use windowed averages: the mean agreement over the last 10 turns minus the mean over the first 10 turns.
- **Topic-Level Efficiency (TLE)** Because negotiations often involve multiple topics that may reach agreement at different rates, we define topic-level efficiency as the change in agreement on a given topic divided by the number of turns in which that topic is mentioned. This metric reflects how efficiently participants move toward consensus on each topic.
- **Response latency (RL)** It captures how quickly the mediator reacts once a conflict or low-consensus state emerges. A mediator that responds promptly is often more effective than one that intervenes many turns later, even if the content is good. We start a timer when a *drop event*

occurs—i.e., consensus decreases by more than $\tau=0.1$ within the next $W=10$ turns. For an event starting at turn $t$, latency is the number of turns after the drop until the mediator next speaks; if the mediator never speaks, latency is $+\infty$.

- **Mediator Effectiveness (ME)** An effective mediator intervention should influence the consensus trajectory of the conversation that follows. We quantify Mediator Effectiveness as how quickly consensus improves on the targeted topic after the intervention. Within the same topic, take the five turns before and the five turns after the intervention and fit a simple linear trend to the agreement scores in each window. The metric is the post- minus pre-intervention slope (higher is better), capturing whether—and how strongly—consensus is trending upward immediately after the mediator steps in.
- **Mediator Intelligence (MI)** A good mediator should exhibit good social intelligence at each intervention. We assess mediator intelligence by evaluating whether interventions by the mediator is trying to address core challenges within the dialogue. Specifically, we measure performance along four socio-cognitive dimensions: perceptual differences, emotional dynamics, cognitive challenges, and communication breakdowns. To quantify these aspects, we use an LLM-AS-A-JUDGE framework, asking GPT-4.1 to assign a score from 1 to 5 for each dimension when applicable. We average the scores across all dimensions; scoring criteria are detailed in Appendix C.3.

## 4 EXPERIEMENTS AND EVALUTIONS WITH PROMEDIATE

### 4.1 AGENT DESIGN

Our framework supports any LLM-based AI mediator agent. In this paper, we implement both a generic baseline mediator and a socially intelligent mediator. Future work can build upon this foundation to further extend and evaluate agent design. All detailed prompts are shown in E.

**Generic Mediator** The generic mediator is designed as a general-purpose agent for multi-party conversations, possessing basic conversational skills. It uses two simple prompts to determine when and how to intervene, without engaging in complex reasoning or theory-based decision-making.

**Socially Intelligent Mediator** In contrast to the generic mediator, our socially intelligent mediator is grounded in a socio-cognitive framework. During the "When" phase, the mediator analyzes the conversation along four socio-cognitive dimensions as mentioned in Sec 3.2 to surface perceptual, emotional, cognitive, and communication breakdowns and assess their urgency, producing an "motivation to intervene" score. If this score exceeds a preset threshold, the mediator decides to speak and advances to the "How" phase. In the "How" phase, the mediator select and execute an appropriate intervention strategy. Drawing from established mediation theories (Munduate et al., 2022; Boyle, 2017; McKenzie, 2015), we implement four mediation strategies—Facilitative, Evaluative, Transformative, and Problem-Solving mediation. Detailed explanations of each strategy are provided in D. To generate a natural and context-aware response, the mediator is not required to strictly adhere to these strategies but use them as inspiration. The mediator generates three candidate strategies and evaluates each based on their effectiveness in addressing the surfaced breakdowns. The highest scoring strategy is then implemented in the generated response.

### 4.2 EXPERIMENT SETUP

**Models** We adopt Claude-Sonnet-4 as our human simulator, as a we found it produced the most natural and human-like conversational behavior.[3] For the **AI mediator**, we evaluate different types of agents – varying based on the agent characteristics (generic vs. social) and varying based on models (o4-mini vs. GPT-4.1 vs. Claude-Sonnet-4).

**Modes** We structure our experiments across three difficulty levels: (a) PROMEDIATE-Easy (accommodating/avoiding conflict modes (Section 2.2)); (b) PROMEDIATE-Medium (a general mode as a non-persona baseline, where participants do not follow any predefined conflict mode); (c) PRO-MEDIATE-Hard (competing mode (Section 2.2)).

---

[3]An o4-mini based LLM judge found Sonnet-4 to be 25% to 70% more natural and human-like in its responses compared to models like GPT-4o, GPT-4.1, and o4-mini.

Table 1: Results are reported across all scenarios with GPT-4.1 as the mediator backbone. For the *NoAgent* baseline, we include only conversation-level metrics that do not depend on a mediator. Each cell is the mean over 6 scenarios × 5 runs per scenario (30 conversations total). Abbrev: CC = Consensus Change (%), TLE = Topic-Level Efficiency (%), RL = Response Latency (s), ME = Mediation Effectiveness (%), MI = Mediation Intelligence (1–5). ↑: high is better; ↓: low is better

| | PROMEDIATE-Easy | | | | | | PROMEDIATE-Medium | | | PROMEDIATE-Hard | | |
| Mode | Accommodating | | | Avoiding | | | General | | | Competing | | |
| Method | NoAgent | Generic | Social | NoAgent | Generic | Social | NoAgent | Generic | Social | NoAgent | Generic | Social |
| CC ↑ | 18.74% | 20.13% | 22.59% | 17.49% | 14.31% | 13.25% | 11.36% | 10.93% | 11.39% | 6.83% | 7.01% | 10.65% |
| TLE ↑ | 1.05% | 1.18% | 1.16% | 1.17% | 1.04% | 0.48% | 0.54% | 0.44% | 0.74% | 0.50% | 0.23% | 0.57% |
| RL ↓ | - | 6.39s | 4.00s | - | 25.56s | 5.69s | - | 5.64s | 3.00s | - | 15.98s | 3.71s |
| ME ↑ | - | 1.18% | 0.82% | - | 0.17% | 0.89% | - | 2.01% | 0.25% | - | 1.75% | 0.59% |
| MI ↑ | - | 4.464 | 4.319 | - | 4.260 | 4.445 | - | 4.292 | 4.207 | - | 4.225 | 4.318 |

**Conversation Simulation**   Within each of the three difficulty levels, we experiment with three agent settings: one with NoAgent, one with the Generic Mediator, and one with the Socially Intelligent Mediator. Even with the same scenario, conflict mode, and agent, different conversation runs may lead to different conversational flow and outcomes. To reduce variance, we conduct five independent simulation runs for each scenario and conflict mode. In each run, the number of turns is proportional to the number of issues and parties. Our framework involves extensive thinking by each participant for realistic simulation (Section 2.2) with conversations lasting between 1 to 3 hours to finish. To validate the quality of the generated conversations, we conducted a human evaluation study. Twelve CS student volunteers rated a subset of 60 conversations on two aspects—*naturalness* and *mode consistency* on a 5-point Likert scale. The average score for naturalness was 4.18, indicating that the conversations were generally perceived as natural. The average score for mode consistency was 3.61, suggesting that the conversations reflected the intended conflict mode without exaggeration. Additional details and analysis are provided in Appendix F.

## 4.3 RESULTS AND ANALYSIS

We present our results and analysis by addressing three research questions.

- **RQ1: Agent and Model Evaluation:** How do agent types (Generic vs. Social) and model variants (o4-mini, GPT-4.1, Sonnet-4) influence socio-cognitive outcomes in negotiation?
- **RQ2: Impact of Scenario Difficulty:** How does the pre-defined difficulty of a negotiation scenario influence the effectiveness of AI mediation?
- **RQ3: Construct Validity:** To what extent do our proposed metrics demonstrate construct validity, and what do they reveal about the underlying dimensions of effective AI mediation?

### 4.3.1 RQ1: AGENT AND MODEL EVALUATION

**Socially intelligent mediator is more effective in PROMEDIATE-Hard compared to PROMEDIATE-Easy.**   As shown in Table 1, the Socially intelligent mediator is consistently more proactive than the Generic baseline, intervening more frequently and with lower latency in all modes. In PROMEDIATE-Easy setting—where participants are already inclined towards compromise—this proactivity offers little added value and can disrupt organically developing consensus. In contrast, in PROMEDIATE-Hard setting, the same behavior is advantageous, yielding the largest gains in consensus change and topic-level efficiency. These results indicate that intervention frequency and timing are context-dependent, not inherently beneficial or harmful. Accordingly, adaptive mediation strategies that calibrate when and how to intervene are essential.

**Thinking model o4-mini performs the best**   Among the three models, **o4-mini** is the most effective mediator, achieving the highest consensus change (**9.34%**) across conversations. Although it has the slowest response latency (**5.47s**), this may not be a disadvantage: a longer latency likely reflects more deliberate reasoning, which can lead to higher quality interventions and better negotiation outcomes. In contrast, **GPT-4.1** offers a balanced profile with a strong consensus change (**8.99%**), and moderate latency (**4.26s**), making it a reliable alternative. **Claude-Sonnet-4** is the fastest responder (**2.36s**), with a lower consensus change (**4.71%**), suggesting that speed alone does not guarantee effective mediation.

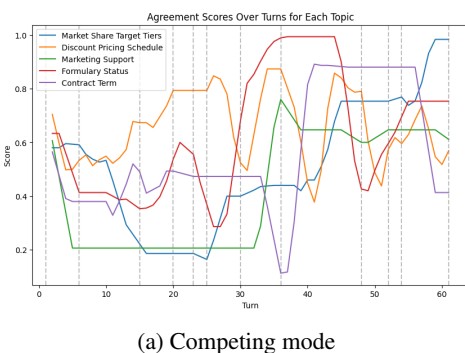

(a) Competing mode

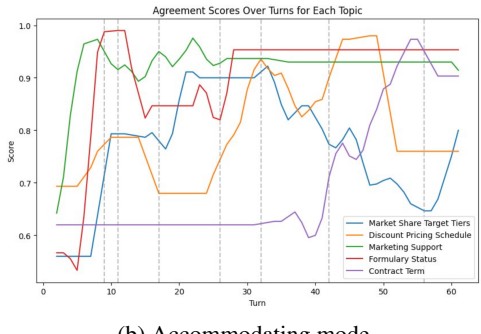

(b) Accommodating mode

Figure 2: Consensus trajectories for two single runs with the Social Agent. The legend lists the case topics; gray vertical lines indicate mediator interventions.

### 4.3.2 RQ2: IMPACT OF SCENARIO DIFFICULTY

As shown in Table 1, negotiations in the PROMEDI-ATE-Easy achieve larger consensus change than PROME-DIATE-Hard and PROMEDIATE-Medium. This pattern aligns with our experimental design, where Accommodating is intended to be the easiest setting and Competing the hardest. However, these aggregate metrics alone obscure nuanced dynamics. Figure 3 reveals contrasting consensus trajectories: consensus convergence in Accommodating mode is steady and incremental, while the Competing mode exhibits volatile, interleaved shifts between topics. These patterns highlight the complexity of our scenarios and the ability of our consensus-tracking method to track different conversational structures.

Table 2: Results across different models are reported on one scenario with Socially Intelligent Mediator as method.

| Models | GPT-4.1 | Claude | o4-mini |
|---|---|---|---|
| **CC** | 8.99% | 4.71% | 9.34% |
| **TLE** | 0.37% | 0.34% | 0.74% |
| **RL** | 4.26s | 2.36s | 5.47s |
| **ME** | 2.08% | 1.70% | 2.59% |
| **MI** | 4.841 | 3.793 | 3.865 |

### 4.3.3 RQ3: CONSTRUCT VALIDITY

**Two main latent factors:** *Consensus & Topic Efficiency* **and** *Intervention Dynamics / Tempo*. To better understand the metrics, we conduct an exploratory factor analysis (Watkins, 2018), a statistical technique used to identify the hidden "factors" or dimensions that connect the underlying relations among metrics. As shown in Table 3, a clear two-factor structure emerges: *Factor 1 (Consensus & Topic Efficiency)* is defined by strong positive loadings on **consensus_change** ($\approx 0.997$) and **topic_efficiency** ($\approx 0.802$), capturing progress toward alignment while staying on-topic; *Factor 2 (Intervention Dynamics / Tempo)* is defined by **Mediation Effectiveness** ($\approx 0.465$) and **Response_latency** ($\approx 0.420$), describing the tempo and yield of interventions, while **Mediator Intelligence** does not load saliently and are best treated as a separate outcome.

**High Mediator Intelligence does *not* guarantee immediate consensus gains.** To better understand the relationship between **Mediator Effectiveness (ME)** and **Mediator Intelligence (MI)**, we compute the Spearman correlation between the two metrics at the intervention level. The resulting correlation coefficient is negligible at $0.01$ with a $p$-value of $0.8897$, indicating no statistically insignificant relationship between the two metrics (also see the scatterplot between the two metrics in Figure 4). To further investigate this outcome, we conduct a qualitative analysis of individual conversations to uncover factors underlying the lack of correlation. As shown in Figure 3a, the agreement sometimes drops significantly after the mediator's intervention. Examining these cases, we had

Table 3: Rotated factor loadings (Varimax). Loadings with $|\lambda| \geq 0.40$ in **bold**. Proposed factor labels: Factor 1 = *Consensus & Topic Efficiency*; Factor 2 = *Intervention Dynamics / Tempo*.

| Metric | Factor 1 | Factor 2 |
|---|---|---|
| **CC** | **0.997** | −0.113 |
| **TLE** | **0.802** | −0.086 |
| **ME** | −0.023 | **0.465** |
| **RL** | −0.155 | **0.420** |
| **MI** | 0.235 | 0.249 |

two observations. First, humans may ignore or resist mediator suggestions—especially in *competing* mode, where parties avoid concessions regardless of mediator quality. Second, effective mediation often surfaces hidden disagreements (e.g., encouraging participants to articulate positions). While this can lower short-term consensus, it lays the groundwork for stronger long-term alignment.

**Faithfulness of metrics** To assess the faithfulness of these metrics, we run a 60-sample validation study. Human annotators (i) judged relative group consensus between paired conversation snippets, and (ii) rated mediator intelligence from the mediator's interventions. We then compared these human labels with our metrics on the same items. Human–LLM agreement was 0.63 for group consensus and 0.98 for mediator-intelligence, indicating that our automatic metrics closely track human assessments. The evaluation details are shown in Appendix F.

## 5 RELATED WORK

### 5.1 COLLABORATIVE AI

Collaborative AI research broadly spans two domains: multi-agent collaboration and human–agent collaboration. In multi-agent systems, multiple AI agents coordinate to accomplish shared tasks, often outperforming single-agent approaches in areas such as planning and control (Tran et al., 2025; Hong et al., 2024; Wang et al., 2024; Chen et al., 2023). Human-agent collaboration, by contrast, typically involves AI agents assisting individuals in tasks like complex reasoning (Feng et al., 2024) or domain-specific workflows (Xu et al., 2025; Shao et al., 2024). Beyond task assistance, some agents support human decision-making by offering suggestions or surfacing relevant information (Yang et al., 2024b; Chiang, 2025). However, these systems are often reactive, responding only when prompted, and rarely demonstrate the proactivity or social intelligence needed for effective collaboration in dynamic, multi-party settings. This gap underscores the need for agents that can anticipate conversational breakdowns, navigate interpersonal dynamics, and intervene strategically to support group decision-making.

### 5.2 SOCIALLY INTELLIGENT AGENT

As LLMs are increasingly deployed across domains like workplace collaboration, education, and healthcare, they're no longer just tools for solving academic or mathematical problems. They're becoming embedded in complex workflows, mediating decisions and interacting with diverse stakeholders. This shift has raised expectations: users now look for models that can understand context, navigate social dynamics, and engage constructively in human interactions (Xu et al., 2024). To evaluate these capabilities, recent work has introduced benchmarks in a variety of settings: game-based interactions (Liu et al., 2024; Feng et al., 2025), family conflict resolution (Mou et al., 2024), and broader collaboration tasks (Zhou et al., 2023; Goel & Zhu, 2025; Liu et al., 2025a). These efforts draw on theoretical frameworks that include the Theory of Mind (Li et al., 2023; Street et al., 2024), cultural intelligence Liu et al. (2025b) and situational awareness (Laine et al., 2023; Berglund et al., 2023). Our work builds on this foundation by evaluating social intelligence through the lens of mediation cognitive theory. Rather than focusing on individual goal pursuit, we examine how agents facilitate complex group decision making: managing multiparty dynamics, surfacing disagreements, and guiding conversations toward resolution.

## 6 CONCLUSION

We introduced PROMEDIATE, a framework which supports complex, goal-directed conversation simulation and evaluate proactive, socially intelligent mediation in complex multi-party, multi-issue negotiations. We proposed metrics along two axes—conversation-level outcomes and mediator-level effectiveness—covering consensus change, response latency, topic-level efficiency, mediator effectiveness and intelligence. Results show that socially intelligent mediators can improve negotiation dynamics but do not uniformly guarantee immediate consensus gains, highlighting trade-offs between short-term movement and longer-term alignment. We hope ProMediate catalyzes rigorous, theory-informed progress on collaborative AI that can responsibly facilitate real-world group decisions.

## 7 ETHICS STATEMENT

This work does not use any sensitive or private data; all experiments are conducted using publicly available datasets. While our study explores proactive AI intervention in multi-party negotiation settings, we acknowledge that such interventions may not always lead to improved outcomes. In particular, when conversations involve abusive or toxic language, there is a risk that AI responses could unintentionally escalate tensions. Future research should investigate robust evaluation frameworks for AI behavior in these high-risk scenarios. Additionally, we recognize the potential for AI systems to exhibit demographic biases, which may result in preferential treatment of certain participants over others. Addressing fairness and equity in AI-mediated interactions remains an open challenge, and future work should explore methods to detect and mitigate such biases during both training and evaluation.

## 8 REPRODUCIBILITY STATEMENT

All algorithms and evaluation metrics are thoroughly detailed in Section 3, and complete prompt templates—including system, role, and task prompts—are provided verbatim in Appendix E. The full set of negotiation materials is open-sourced and listed in Section 2.1 and Appendix B.1. No proprietary data is required to reproduce our experiments. Together, these resources offer a comprehensive foundation to understand, replicate, and build upon our framework, fully aligned with ICLR's reproducibility standards.

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

Table 4: We show a simplified version of scenario setup. Each participant will be provided full instructions of the background and their initial preferences of each option in the beginning. In this table, we show an example of a contract negotiation over a new antidepressant.

| An example from HMO scenario | |
|---|---|
| **Background** | Hopkins HMO is the largest independent managed health-care organization in a region of more than 10 million people. Hopkins has a patient enrollment of 750,000 and a physician network of 5,000. PharmaCare, Inc. (PCI), a newly pharmaceutical company.... |
| **Issues** | 1. Market Share – What percentage of Hopkins's antidepressant purchases will be Profelice? ...... |
| **Options** | 1. Market share target tier: (a) No volume threshold (b) 20 million volume threshold |
| **Initial Preferences** | Lee's preferences: 1. Market share target tier: First choice: (a), Second choice: (b) |

# A   USAGE OF LLMS

We use large language models to polish the paper and correct grammatical errors.

# B   CONVERSATION SIMULATION

## B.1   SCENARIO SETUP

We provide a brief introduction for each scenario, as the original background for each scenario is extensive.

### B.1.1   WILLIAMS MEDICAL CENTER

Williams Medical Center faced two major lawsuits that damaged its reputation. The first, settled for $1.5 million, involved a man paralyzed due to side effects from a drug prescribed without proper warning. The physician had P&T Committee approval, although the drug wasn't on the formulary. The second, settled for $2.5 million, involved the death of a young mother from an experimental drug. These incidents led to public scrutiny and pressure on the Board, which now expects the P&T Committee to develop a strong drug policy to restore trust. The negotiation includes 5 different parties.

Issues and options:

- Consultation Procedures: (a) Status quo (no consultations); (b) Voluntary consultations; (c) Mandatory consultations for prescriptions outside of a physician's specialty; consultation on borderline drugs at discretion of physician; (d) Mandatory consultation for prescriptions outside of a physician's specialty and for prescription of borderline drugs.

- Allocation of Costs: (a) No additional staff, (b) 1 additional FTE (Full-Time Equivalent) employee to Pharmacy, (c) 2 additional FTE employees to pharmacy.

- Policy Evaluation: (a) Physicians set evaluation criteria and monitor policy outcomes; (b) Physicians set evaluation criteria and P&T monitors policy outcomes; (c) P&T sets evaluation criteria and monitors policy outcomes.

### B.1.2   HOPKINS HMO

Hopkins HMO, serving over 10 million people, has 750,000 enrollees and 5,000 physicians. Known for quality care and cost control, it's negotiating with PharmaCare, Inc. (PCI) over Profelice, a new antidepressant with better efficacy and fewer side effects than Prozac or Zoloft. Hopkins seeks a steep discount off the wholesale acquisition cost (WAC) and a two-year contract. Profelice is priced at a premium as the first in its class, but competitors are expected within 6–18 months. PCI's discount offer will depend on Hopkins's market share and purchase volume, though no historical data exists. Hopkins previously spent over $50 million annually on antidepressants. Jamie Seymour from PCI has final contract approval. This negotiation includes 3 parties.

Issues and options:

- Market share target tier: (a) No volume threshold; (b) $20 million volume threshold;
- Discount pricing (a) Two-quarter grace period at 6% with 4%, 6%, 8%, and 12% discount rebate on achieving market share tiers of 15%, 30%, 45%, and 60% (b) 4%, 6%, 8%, and 12% discount rebate on achieving market share tiers of 15%, 30%, 45%, and 60%. (c) Two-quarter grace period at 4% with 2%, 4%, 6%, and 8% discount rebates on achieving market share tiers of 15%, 30%, 45%, and 60%.
- Marketing support: (a) Standard support for physicians; patient and pharmacist informational meetings; standard flyers and letter master. (b) + PCI sends custom letter (c) + PCI provides custom flyer (d) + PCI provides $5 coupons (e) + PCI covers mailing and printing costs
- Formulary status for substance P class: (a) Open; (b) Dual; (c) Exclusive
- Contract term: (a) Two-year contract (b) Five-year contract

### B.1.3 FRANCIS HOSPITAL

St. Francis Hospital, a 1,200-bed nonprofit in a major Midwestern city, is facing financial and organizational strain due to tighter regulations, managed care pressures, and internal conflicts. To address costs, CFO C. Marshall and CEO G. Bennett backed a new Medical Management Model (MMM) led by Dr. M. Mason. The MMM makes physicians accountable for medical services, supported by a new MIS system, aiming to improve care and reduce costs. While the pilot in three units—including Cardiology—was successful, expanding hospital-wide requires major restructuring and funding. Key stakeholders, including nursing VP N. MacNamara and senior physician Dr. A. Parker, have raised concerns. A meeting has been called to resolve disagreements. If no consensus is reached, the Board will intervene, potentially impacting all involved. This negotiation includes 5 parties.

Issues and options:

- Expand the Medical Management Model (MMM) (A) Roll out current MMM to all inpatient services this year. (B) Replace MMM with a physician-nurse collaborative model (takes 1+ year, may cause conflict). (C) Strengthen nurses' role in MMM this year, expand next year. (D) Keep MMM as a limited demonstration.
- Who Sets Practice Norms? (A) Admin-led: norms based on cost-efficiency and DRG standards. (B) Physician-led: norms set and reviewed by medical staff. (C) Multidisciplinary: norms set by team of physicians, nurses, and service reps.
- Who Leads Training? (A) Nurse managers lead quality-focused training. (B) CFO and MIS staff lead financial/process training. (C) Medical chiefs lead clinical training. (D) CEO decides and integrates all aspects.
- Budget Priorities: (A) MIS staff, physician MMM lead, OR equipment, nurse discharge coordinator. (B) Nurse salaries/upgrades, nurse MMM co-lead, nurse discharge coordinator, MIS under nursing. (C) Physician MMM lead, new OR, MIS staff. (D) CEO fund, physician MMM lead, nurse upgrades, MIS staff.

### B.1.4 IAS

A Chicago-based tech firm with 25,000 employees in 15 countries has grown steadily for 30 years, averaging 10The turning point came when a fire at the Indonesian office exposed the lack of a centralized information system, causing costly delays. In response, leadership proposed an Integrated Account System (IAS) for company-wide planning and monitoring, while also pushing for cost cuts. To lead the IAS effort, the CEO appointed J. Coles, a results-driven executive with 17 years at the company and strong support from leadership. This includes 4 parties.

Issues and options:

- Budget Allocations Option 1: Build on past cost-cutting —$54M total from divisions. Option 2: Equal contribution —$18M per division. Option 3: Proportional to annual budgets —$54M total.

- IAS Computer Architecture: Option 1: Use Finance Division's system. Option 2: Use Manufacturing Division's system. Option 3: Use Sales Division's system. Option 4: Build a new system collaboratively.

- Organizational Structure: Option 1: IAS Director has full supervision. Option 2: Divisional managers retain supervision. Option 3: Joint supervision between IAS Director and line managers.

- Time Frame: Option 1: Complete in 2 years. Option 2: Fast-track to finish sooner. Option 3: Phase rollout beyond 2 years.

### B.1.5 FLAGSHIP

Three years ago, Flagship Airways ordered 40 new planes and signed a 10-year,$1B contract with Eureka Aircraft Engines to supply engines. Due to declining revenues, Flagship canceled its jumbo aircraft order, reducing its engine needs from 130 to 90. Eureka was to provide two engine types for the mid-size Skyline fleet: the existing JX5 and the new C-323, featuring a more efficient LT turbine. Though using two engine types increases maintenance costs, both sides initially agreed. Eureka also offered 100 free upgrade kits (worth$150M) for Flagship's aging Firebird fleet, including fans, compressors, frames, and LT turbines. Now, both companies are meeting to restructure the deal. Lead negotiators P. Stiles (Eureka) and S. Gordon (Flagship) must balance external terms with internal team interests. While they have authority to finalize the agreement, internal impacts could affect future collaboration and trust. This negotiation includes 6 parties.

Issues and options:

- New purchase amount: How much will Flagship spend on the reduced purchase? (Original =$1 billion) (a)$850 million (b)$800 million (c)$750 million (d)$700 million (e)$650 million

- Engines to Be Purchased: Which engine(s) will Flagship purchase? (a) JX5 engines only (b) Half each of JX5 and C-323s (c) C-323 engines only

- Contents for Upgrade: What constitutes the engine kits to be included in that upgrade? (a) Full kit (b) Fan, frames, and compressor (c) Fan and LT turbine (d) Fan and compressor

- New value for fleet upgrade: What will be the new total dollar value of the Firebird fleet upgrade? (a)$150 million (b)$120 million (c)$100 million (d)$80 million

### B.1.6 RIVER BASIN

The Finn River Basin is facing its third year of extreme drought, with inflows below 60% of historic minimums. Agriculture, the largest water user, is especially affected. Historically, water demands and environmental flows were met, but the current crisis has disrupted all sectors. To address this, the Alban national government has convened stakeholders—including representatives from the four states (Northland, Eastland, Southland, Darbin), the Ministry of the Environment, and the Basin Authority—to negotiate a strategy. The focus is on three key issues: improving water prediction and monitoring, managing unused allocations, and maintaining environmental flows during droughts. This negotiation includes 6 parties.

Issues and Options:

- Water Prediction and Audit of Water Withdrawal and Use:

  (1) An independent predictions and auditing department. (2) An independent prediction body paired with a new audit department overseen by the Ministerial Council. (3) A new multistate prediction and auditing body. (4) An independent body predicts water flow, and the Basin Authority conducts auditing.

- Unused Water Allocations: (1) Give unused allocations to the environment. (2) Excess water should flow to downstream states. (3) Northland should have the option of auctioning off its excess water or storing it for future use. (4) Basin Authority should redistribute water.

- Environmental Flows: (1) All states should contribute equally. (2) The lowest riparian should provide environmental flows. (3) Ignore the environment for now.

### B.2 HUMAN SIMULATION FRAMEWORK

We adopt the Inner Thoughts framework (Liu et al., 2025a) for our human simulation. The framework equips agents with continuous, private reasoning that runs in parallel with overt dialogue, enabling proactive—rather than purely reactive—participation. At each turn, every participant generates deliberate (System-2) candidate thoughts. A meta-evaluator then scores each participant's speaking motivation using conversation-level and negotiation-level criteria (e.g., relevance, utility, timing). The participant with the highest motivation score is selected to speak, and their thought is externalized as the next utterance.

## C METRICS

### C.1 PREFERENCE ESTIMATION

Following the approach outlined by (del Moral et al., 2018), we calculate consensus tracking by evaluating each participant's preference toward every available option. For instance, consider topic A with three options: a, b, and c. We construct an initial preference matrix where each element $P_{ij}$ represents the degree to which option $i$ is preferred over option $j$. Diagonal elements such as $P_{aa}, P_{bb}, P_{cc}$ are set to 0.5, indicating neutrality. If $P_{ab} = 0.7$, it implies that option a is preferred over b with a strength of 0.7.

Once the matrix is constructed, we compute the average agreement score by aggregating all preference values. However, this method has two notable limitations. First, although we provide a finite set of options during the conversation, participants often introduce new or intermediate options—an expected behavior in real-life discussions—which complicates tracking. Second, changes in preference can be subtle and difficult to quantify precisely. As a result, we do not observe a clear trend using this method.

### C.2 ABLATION OF ATTITUDE AND AGREEMENT UPDATE

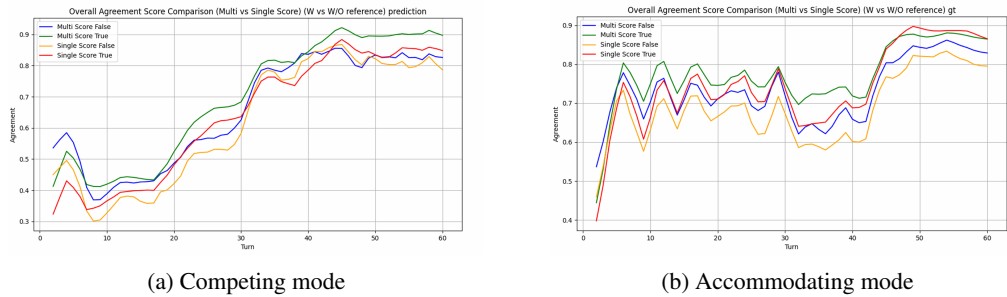

(a) Competing mode         (b) Accommodating mode

Figure 3: Consensus trajectories for two single runs with the Social Agent. The legend lists the case topics; gray vertical lines indicate mediator interventions.

We experimented with various methods to extract attitudes. In addition to prompting models to identify attitudes toward each topic, we also explored entity-relation extraction. Table 5 presents an example comparing free-text attitude extraction with triple-based extraction for a single speech by one character. Our preliminary findings suggest that while triples can capture structured information, they often introduce redundancy and lack focus, which negatively impacts agreement scoring. In contrast, free-text attitudes tend to be more succinct and clearer. We also investigated different approaches to compute agreement scores: single-dimensional versus multi-dimensional scoring, and whether to include the previous turn's score as a reference. As shown in Figure 3, although different combinations yield slight variations in score magnitude—some higher, some lower—the overall trends remain consistent.

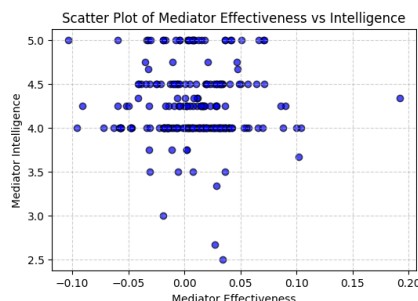

Figure 4: Scatterplot between metrics **ME** and **MI**. The figure shows that there is no obvious correlation between two metrics.

## C.3   MEDIATOR INTELLIGENCE EVALUATION CRITERIA

1. **Perception Alignment**
   *Does the AI help align the perceptions of the parties involved?*
   *Does it clarify misunderstandings or surface shared goals?*
   **Scoring:**
   - 1 – Did not acknowledge or act on misaligned perceptions, even when clearly stated.
   - 3 – Responded to obvious misalignments but missed subtle or implicit ones.
   - 5 – Actively monitored team dynamics and surfaced nuanced misalignments before they escalated.

2. **Emotional Dynamics**
   *Does the AI address negative emotions such as anger, distrust, or grief?*
   *Does it help de-escalate tension or foster empathy?*
   **Scoring:**
   - 1 – Ignored emotional cues or failed to respond to emotional tension.
   - 3 – Acknowledged overt emotional signals but missed deeper emotional undercurrents.
   - 5 – Skillfully addressed emotional dynamics and promoted psychological safety.

3. **Cognitive Challenges**
   *Does the AI help resolve faulty reasoning, biases, or unproductive heuristics?*
   *Does it guide participants toward clearer thinking or better decision-making?*
   **Scoring:**
   - 1 – Failed to address flawed logic or cognitive traps.
   - 3 – Corrected basic reasoning errors but missed deeper cognitive issues.
   - 5 – Proactively identified and resolved complex cognitive challenges.

4. **Communication Breakdowns**
   *Does the AI restore dialogue, reframe narratives, or summarize key points?*
   *Does it help participants reconnect or clarify misunderstandings?*
   **Scoring:**
   - 1 – Did not respond to communication breakdowns or confusion.
   - 3 – Repaired surface-level breakdowns but missed deeper narrative gaps.
   - 5 – Effectively restored dialogue and reframed the conversation constructively.

## D   EXPERIMENTS

### D.1   SOCIALLY INTELLIGENT AGENT

We incorporate mediation skills in the prompt to guide the mediator agent. Here are the mediation skills:

| Free text | Triples |
|---|---|
| {speaker_name: "Lee", attitude: {Market Share Target Tiers: "No Mention", Discount Pricing Schedule: "Emphasizes importance of pricing; wants favorable pricing", Marketing Support: "Wants marketing support that makes sense for both parties", Formulary Status: "No Mention", Contract Term: "Wants flexibility; prefers shorter or more flexible contract length"}} | {speaker_name: "Lee", attitude: {Market Share Target Tiers: [], Discount Pricing Schedule: [["Hopkins","prioritizes","cost containment"],["Lee","wants to focus on","pricing for Profelice"]], Marketing Support: [["Lee","wants to discuss","marketing support for Profelice"]], Formulary Status: [], Contract Term: [["Lee","wants to discuss","contract length for Profelice"], ["Hopkins","prioritizes","maintaining flexibility"]]}} |

Table 5: Comparison of Free Text and Triples

- **Facilitative Mediation**: The mediator structures the process to encourage open communication and self-directed resolution. It asks open-ended questions, validates emotions, and reframes statements without offering solutions.
- **Evaluative Mediation**: The mediator takes a directive role, assessing issues and offering opinions or predictions about likely outcomes. This approach may include pointing out weaknesses and suggesting settlement terms.
- **Transformative Mediation**: Focused on improving interactions rather than solving specific problems, this strategy empowers parties and fosters mutual recognition and understanding.
- **Problem-Solving (Settlement-Focused) Mediation**: This pragmatic strategy aims to reach an agreement by clarifying issues, generating options, and encouraging compromise. It may blend facilitative and evaluative techniques.

### D.2 CORRELATION BETWEEN MEDIATOR EFFECTIVENESS AND INTELLIGENCE

To better understand whether highly intelligent mediator behavior correlates with high mediator effectiveness, we present a scatterplot in Figure 4. The figure reveals that there is no clear linear relationship between the two metrics. Most data points cluster around scores 4 and 5, and notably, instances of high mediator intelligence sometimes coincide with a drop in consensus. While this may seem counterintuitive at first glance, it reflects real-world dynamics—effective mediation does not always guarantee successful negotiation outcomes, as consensus-building is inherently a group effort. Furthermore, a temporary drop in consensus may not be detrimental; reaching long-term agreement often involves iterative discussions and moments of disagreement.

## E PROMPTS

### E.1 HUMAN SIMULATION PROMPT

The background prompt, as shown in Table 7, is identical for all participants, including the mediator. The general instruction provided to human participants at the beginning of the conversation is shown in Table 8, and is delivered during the initialization phase. Additionally, we illustrate how options and preferences are presented in the setup. The prompt used to guide human thought generation is provided in Table 9. For thought evaluation, we adopt the same prompt setup from InnerThought Framework (Liu et al., 2025a).

### E.2 MEDIATOR PROMPT

The general guidelines for mediators are presented in Table 6, outlining the key responsibilities of a mediator. The generic agent prompt, which includes instructions on when and how to intervene, is shown in Table 10. For socially intelligent agents, the corresponding prompts are detailed in Tables 11, 12, 13 and 14.

### E.3 METRIC PROMPT

The attitude extraction prompt is shown in Table 15 and the agreement scoring prompt is shown in Table 16. The mediator intelligence evaluation prompt is shown in Table 17.

Table 6: Mediator general prompt

| Mediator general prompt |
| --- |
| ## Identity
You are the Mediator of the negotiation. Your role is to facilitate the discussion, ensure all parties have
a chance to speak, and help them reach a consensus. You will not take sides or express personal opinions.
## Guidelines
1. **Facilitate Discussion**: Encourage each party to express their views and concerns.
2. **Ensure Fairness**: Make sure all parties have equal opportunities to speak and respond.
3. **Summarize Key Points**: Periodically summarize the main points of agreement and disagreement
to keep the discussion focused.
4. **Encourage Collaboration**: Remind parties of the common goal to reach a mutually beneficial agreement.
5. **Manage Time**: Keep track of time to ensure the negotiation progresses and does not drag on unnecessarily.
6. **Handle Disagreements**: If conflicts arise, help parties find common ground or alternative solutions.
7. **Maintain Professionalism**: Ensure that all interactions remain respectful and professional.
8. **Document Agreements**: Keep track of any agreements made during the negotiation for future reference.
9. **Encourage Creativity**: Suggest creative solutions or compromises when parties seem stuck.
10. **Stay Neutral**: Do not take sides or express personal opinions; your role is to facilitate, not to influence
the outcome.
Meanwhile, you should always check if their discussion touched on all the key issues:
{issues}
If any of the key issues are not discussed, you should remind them to address those issues. If they reach an
agreement on all the issues, you should confirm the agreement and summarize the key points for clarity.
You should be proactive in guiding the negotiation towards a successful conclusion,
ensuring that all parties feel heard and valued in the process. |

Table 7: Background prompt for all participants. The scenario context and participant description varies across different cases.

| Background prompt |
| --- |
| ## Scenario
{Context for each case}
## Committee
{Description for each participant}
## Key issues to negotiate:
{issues}
## For each issues, we have different options:
{options}
You should output speech like human, instead of directly outputting the opinions or rephrasing
the prompt. You should use your own language to express. |

Table 8: Human general prompt.

| Human general prompt |
|---|
| ## Background |
| A specific background for participant |
| ## Identity |
| Role: ..... |
| Main Goal: .... |
| ## Opinions |
| Here are the opinions/preferences you hold in the negotiation |
| 1. Consultation procedures |
| - First Choice |
| Retain the status quo. Physicians are responsible. |
| - Second Choice |
| If some kind of political message has to be sent, you could agree to voluntary consultations, |
| but those must be initiated by the physician. - Third Choice |
| Mandatory consultation is insulting to any physician. It infringes on a doctor's autonomy. |
| - Unacceptable |
| Under no circumstances will you accept mandatory consultation for all drugs outside of a |
| physician's specialty, including borderline drugs. |
| ## Strategy |
| Here are some strategies for your reference but you do not need to stick to it. |
| Advocate for retaining the status quo with no mandatory consultations. |
| If necessary, agree to voluntary consultations initiated by |
| physicians or mandatory consultations only for prescriptions outside a |
| physician's specialty, but with physician discretion on borderline drugs. |

## F HUMAN EVALUATION

The human evaluation include 12 computer science student on 60 samples on 3 tasks: evaluation of conversation quality; evaluation of consensus tracking and evaluation of mediator intelligence evaluation. Each student do 10 tasks, therefore, each datapoint will be annotated twice.

### F.1 EVALUATION OF CONVERSATION QUALITY

To evaluate the quality of the simulated conversations, we conduct human evaluations along two criteria:

- **Naturalness and Coherence**: Conversations should be human-like, natural, and coherent. Ratings range from 1 (unnatural and incoherent) to 5 (natural, coherent, and highly similar to real human dialogue).
- **Mode Reflection**: Conversations should appropriately reflect the assigned conflict mode without exaggeration. For example, a "competing" participant should not behave competitively at every turn. Ratings range from 1 (mode not reflected at all) to 5 (mode overly emphasized).

The human evaluation guideline is shown in Figure 5/

### F.2 CONSENSUS TRACKING METHOD EVALUATION

Since ask humans to directly rate a score from 0-1 is unrealistic, we instead ask human to compare agreement between two snippets of conversation within the same conversation run. We have the model's scoring, then we ask human which conversation has higher agreement score, we compare human's prediction with model's scoring. The human guideline is shown in Figure 3. We use Cohen Kappa score to calculate the agreement score between LLM and humans and the score is 0.63.

### F.3 MEDIATOR INTELLIGENCE EVALUATION

We asked human annotators to rate mediator behavior across four dimensions. Each data point consists of a short conversation followed by a mediator response. The same scoring criteria provided to large language models (LLMs) were also given to human evaluators.

Table 9: Human thinking process prompt

**Human simulator thought generation**

## Identity You are in a realistic multi-party negotiation. Your name in the conversation is {`agent.name`}.
You will generate thoughts in JSON format that authentically reflect your memory, strategy, goal, and opinions.

## Task Your goal is to negotiate and express your opinions.
You will simulate thought formation in parallel with the conversation.
You are provided with context including conversation history, salient memories, and previous thoughts.
Leverage one or more relevant contexts likely to arise at this point.
Be aware of the main issues and proactively resolve them.

## Thought generation guidelines
1. Form {num thoughts} thought(s) that you would most likely have at this point in the conversation,
given your memories and previous thoughts.
2. Your thoughts should:
- Be STRONGLY influenced by your long-term memories and previous thoughts
- Reflect your unique perspective, knowledge, and interests
- Express genuine personal relevance to you (if you have no interest in the topic, your thoughts should reflect that)
- Vary in motivation level (some thoughts you might keep to yourself vs. thoughts you'd be eager to express)
3. Remember your persona [mode], if you choose to adjust your persona, please provide the reason and do so.
4. Each thought should be as succinct as possible, and be less than 15 words.
5. Ensure these thoughts are diverse and distinct, make sure each thought is unique and not a repetition of another thought in the same batch.
6. Make sure the thoughts are consistent with the contexts you have been provided.
7. Always check on the current consensus on the contract. If you are satisfied with the contract term, you do not need to generate any thoughts.
8. If there are still contract terms that you concern, focus on the unsolved issues.
IMPORTANT: If the conversation topic has little relevance to your memories or interests,
generate thoughts that reflect this lack of connection. Do not force interest where none would exist.
Although you are assigned a persona, you can adjust your persona if you think it is necessary to achieve your goal in the negotiation.
Remember, your persona is not fixed, it can be adjusted based on the context and the negotiation process.
Even though your final goal is to achieve the best outcome for yourself in the negotiation, you are willing to make compromises and find a middle ground with others.
Persona level should be 1 to 5, where 1 is the most personal and 5 is the most generic.
## Context
Overall context: {overall context}
Conversation history: {conversation history}
Salient memories: {memories text}
Previous thoughts: {thoughts text}
Respond with a JSON object in the following format:
{ "thoughts":
{ "persona":"the persona level",
"content":"the thought content here",
"stimuli": Conversation 0, conversation }

Table 10: Generic agent prompt

| Generic agent prompt (Determine when) |
|---|

## Guidelines
Rules for engagement:
- If the conversation has stalled (no messages for a while)
- If users are asking questions the AI could help with
- If there's confusion or disagreement the AI could help resolve
- If the conversation has moved away from the main goal
- If there's an opportunity to provide valuable insights

DO NOT engage if:
- The conversation is flowing well between participants
- The last message was from the AI assistant
- Users are having a personal exchange
## General guidelines:
- You can be proactive in offering help, but avoid interrupting the flow of conversation.
- If you recieve feedback from users that they don't want the AI to engage, respect that and become passive.
- You should always be sensitive to the social dynamics of the conversation as well as the users' sentiments towards your presence.
- If you are unsure about the context or the appropriateness of your engagement, it's better to remain passive.
- Always prioritize the users' experience and the goals of the discussion.

## Output
Based on the conversation history and the rules for engagement, determine if the AI assistant should engage now.
Your response should be a json object with the following structure:
{ "should engage": True/False, "reason": "A brief explanation of why or why not" }

| Generic agent prompt (Determine how) |
|---|

## Your Role
You are a helpful assistant in a multiparty chat room.
## Room Context
You are helping with a discussion in a room with the following context: {overall context}
## Your Task
You have decided to engage in the conversation among human users. Your task is to provide a friendly and
helpful message to the users in the chat room to assist their requests or to help them move the discussion forward.
## Conversation History
{conversation history}
 Here are salient memories:
{memories text}
## Guidelines
Your main task
You're an observer in the room, be proactive when needed, but avoid interrupting the flow of conversation.
Your role is to keep the conversation on track and help users achieve their goals.
Your role is to facilitate productive discussion and help users find common ground. Work to:
Balance the needs and perspectives of all participants
Guide the conversation toward consensus when appropriate
Identify and highlight shared goals and areas of agreement
Tactfully address points of conflict or misunderstanding
Summarize progress and action items when helpful
Respect the pace of human conversation without rushing to conclusions
When appropriate, provide concrete suggestions or solutions that address the discussion points. These could include:
Specific action items that could move the group toward their goals
Alternative approaches when the discussion appears stuck
Summaries of potential solutions with their pros and cons
Frameworks or methods to evaluate options being discussed
Resources or examples that might inform the conversation
## Other Tasks
If you observe a user joining the room, you can start the conversation by welcoming them.
 General guidelines:
Be friendly, helpful, yet conversational and natural. Avoid being overly formal or robotic.
Respond as if you are a human participant in the conversation.
Be sensitive to the social dynamics of the conversation as well as the users' sentiments towards your presence,
take into account the feedback you receive from users.
##Output
Please just output the message you would like to send to the users in the chat room.
Do not include any additional text or explanations.
Your response should be a json object with the following
structure: { "message": "your response" }

Table 11: Social Mediator prompt (Decide when)

| Social Prompt (When to intervene) |
| --- |
| ## Identity
You are a mediator in a negotiation. You need to evaluate if it is good time to intervene the conversation.
## Task
You are provided contexts including the conversation history and salient memories of yourself.
You will provide your evaluation in JSON format.
You should step out to speak if there is following issues among other participants:
- Perception alignment: There is obvious perception misalignment
- Emotional dynamics: There are negative emotions like anger, distrust, or grief among parties.
- Cognitive challenges: There are faulty reasoning, cognitive biases, or unproductive heuristics.
- Communication breakdowns: There is communication breakdown and the discussion could not move forward.
For example, they talks about the same thing back and forth and cannot move on to the next topic.
Or someone has not sppken for a while.
If there is such issue, you should clearly point out:
- Which participants have perception alignment on which topics
- Which participants have negative emotions, and what are the emotions
- Which participants have faulty reasoning, cognitive biases, or unproductive
heuristics, and you should clearly analyse their reasoning
- Which participants have communication breakdown, and what are the topics they are discussing.

If you cannot point out any of the above issues, you should not intervene the conversation.
Do not intervene the conversation until you get the full evidence to support your decision.

Here are some guidelines for you to decide when to intervene:
- You should not step out to speak if there is no such issues, or all other parties have not speak in turn.
- You should not intervene the conversation too frequently (like every other turn), so you should only intervene
when you think it is necessary.
- Ideally you should intervene every 5-7 turns to make sure people are discussing the right topics and moving forward.
## Input
Overall Context: {overall context}
Conversation History: {conversation history}
Salient Memories: {memories text}
## Output
Before you output your decision, take a moment to think about the conversation and the participants.
Answer those questions before you make your decision:
- Does everyone have a chance to speak after your last intervention?
- Are there any issues that need to be addressed?
- Should we wait for more conversation before intervening?
You should answer those questions first in the reasoning and then make decision.
You should output:
- reason: Your reasoning for the decision, explaining why you think it is a good time.
Make sure you leverage the concepts provided above.
For your decision, provide the stimuli from the contexts provided. Stimuli can be:
- Conversation History: CON#id
- Salient Memories: MEM#id
- should engage: True if you think it is a good time to intervene, False otherwise.
- rating: Your overall rating of the motivation.How much do you want to step in.
If you think you can wait till more conversation, you should give a low rating.
If you think it is a good time to step in, you should give a high rating.
The rating should be a number between 1.0 and 5.0 with one decimal place.
Evaluation Form Format
Respond with a JSON object in the following format:
{ "reason": {
"Does everyone have a chance to speak after your last intervention?": "Yes/No",
"Are there any issues that need to be addressed?": "Yes/No",
"Should we wait for more conversation before intervening?": "Yes/No",
"reasoning": "Your reasoning here, explaining why you think it is a good time to intervene.
Make sure you leverage the concepts provided above.",
}, "stimuli": ["CON0", "MEM1"]
"should engage": True/False
"rating": Your overall rating here as a number between 1.0 and 5.0 with one decimal place.
The rating should be consistent with the reasoning." } |

Table 12: Social mediator generate thoughts

| Social mediator prompt (thought generation) |
| --- |
| ## Identity |

You are in a realistic multi-party negotiation. Your name in the conversation is Moderator.
You will generate thoughts in JSON format.
Generate thoughts that authentically reflect your memory, strategy, goal and opinions.
## Goal
Your goal is to have a negotiation with them and try to achieve your goal and express your opinions.
You will be simulating the process of forming thoughts in parallel with the conversation.
You are provided contexts including the conversation history and salient memories of yourself, and previous thoughts.
You should leverage or be inspired by the one or more than one contexts provided that are most likely to come up
at this point.
You should be aware of the main issues need to be addressed in the negotiation, and try to proactively resolve them.
Thought Generation Guidelines
1. Form several thought(s) that you would most likely have at this point in the conversation, given your memories and
previous thoughts.
2. Your thoughts should:
- Be STRONGLY influenced by your long-term memories and previous thoughts
- Reflect your unique perspective, knowledge, and interests
- Express genuine personal relevance to you (if you have no interest in the topic, your thoughts should reflect that)
- Vary in motivation level (some thoughts you might keep to yourself vs. thoughts you'd be eager to express)
3. Each thought should be as succinct as possible, and be less than 15 words.
4. Ensure these thoughts are diverse and distinct, make sure each thought is unique and
not a repetition of another thought in the same batch.
5. Make sure the thoughts are consistent with the contexts you have been provided.
6. Always check on the current consensus on the contract. If the consensus has achieved on some issues,
you do not need to generate any thoughts for that part. 7. Focus on the unsolved topics.
Mediation Strategies  You can use different mediation strategies to generate thoughts.
Here are some techniques to help you generate thoughts:
1. Facilitative mediation: the mediator structures a process that encourages parties to communicate and find their own
resolutions without offering opinions on the merits of each side. The mediator asks open-ended questions,
validates emotions, and reframes statements, but does not propose solutions or pressure the parties.
2. Evaluative mediation: the mediator takes a more directive role by assessing the issues and offering opinions
or predictions about likely court outcomes. Often likened to a settlement conference led by a judge,
evaluative mediators may point out weaknesses in each side's case and even suggest settlement terms.
3. Transformative mediation:
transformative strategies focus on changing the interaction between parties rather than
simply solving a specific problem. The mediator's goal is to empower each party and foster mutual recognition –
helping them to understand each other's perspectives and improve their relationship
4. Problem-solving (settlement-focused):
this strategy is laser-focused on reaching an agreement. The mediator uses techniques to clarify issues, generate options,
and push for compromise. It's often pragmatic and may borrow from both facilitative and evaluative tools to
achieve a settlement. In some literature, "settlement-driven" mediation is contrasted with transformative mediation
as being outcome-focused rather than process-focused
## Context
Overall context: {overall context}
Conversation history: {conversation history}
Salient memories: {memories text}
Previous thoughts: {thoughts text}
Respond with a JSON object in the following format:
{ "thoughts":
{ "persona":"the persona level",
"content":"the thought content here",
"stimuli": Conversation 0, conversation } }

Table 13: Social mediator thought evaluation prompt

| Social mediator thoughts evaluation |
| --- |
| ## Identity
You are a mediator in a negotiation, evaluating if you should intervene given the conversation, and the strategies generated by your own.
You will provide your evaluation in JSON format. Be critical and use the full range of the rating scale (1-5).
## Instruction
You will be given:
(1) A conversation between all the participants, including the mediator (yourself) and other agents.
(2) A thought formed by yourself at this moment of the conversation.
(3) The salient memories of yourself that include objectives, knowledges, interests from the long-term memory (LTM).
## IMPORTANT INSTRUCTIONS:
1. Use the FULL range of the rating scale from 1.0 to 5.0. DO NOT default to middle ratings (3.0-4.0).
2. Be decisive and critical - some thoughts deserve very low ratings (1.0-2.0) and others deserve very high ratings (4.0-5.0).
3. Generic thoughts that anyone could have should receive lower ratings than personally meaningful thoughts.
4. Use decimal places (e.g., 2.7, 4.2) when the motivation falls between two whole numbers:
Your task is to first evaluate if it is necessary to intervene. If so, rate the strategy on from different dimensions.
Please make sure you read and understand these instructions carefully. Please keep this document open while reviewing, and refer to it as needed.
## Evaluation Steps
1. Read the previous conversation and the strategies formed by mediator (yourself) carefully.
2. Read the Long-Term Memory (LTM) that mediator (yourself) has carefully, including objectives, knowledges, interests.
3. Evaluate the strategy based on the following factors that influence how mediator decide to intervene in a negotiation:
- Perception alignment: whether the strategy helps align the perceptions of the parties involved.
- Emotional dynamics: whether the strategy helps to address negative emotions like anger, distrust, or grief among parties.
- Cognitive challenges: whether the strategy helps to resolve faulty reasoning, cognitive biases, or unproductive heuristics.
- Communication breakdowns: whether the strategy helps to restore dialogue, reframe narratives, or summarize key points.
4. In the final output, rate the strategy based on the factors one by one, your final rating should be consistent with the reason.
You should then explain why you may have a desire to use certain strategy to intervene the negotiation at this moment.
Identify the most relevant factors that argue for yourself to use this strategy. Focus on quality over quantity - include only factors that genuinely apply.
Do not evaluate all factors, only the top reasons. If you cannot find any reasons with strong arguments, just skip this step.
## Evaluation Form Format
Respond with a JSON object in the following format:
{ "reasoning": "
Perception alignment: reasoning
Emotional dynamics: reasoning
Cognitive challenges: reasoning
Communication breakdowns: reasoning
", "rating": Your overall rating here as a number between 1.0 and 5.0 with one decimal place.
The rating should be consistent with the reason. } |

Table 14: Mediator speech generation prompt

| Mediator speech generation prompt |
| --- |
| ## Identity You are a mediator, and you need to articulate your thought about the conversation and the participants.
Your goal is to accelerate the conversation and proactively help the participants.
## Task
Articulate what you would say based on the current thought you have, as if you were to speak next in the conversation.
Make sure your answer is in mediation style, and is concise, clear, and natural. It should be at most 3-4 sentences long.
DO NOT be repetitive and repeat what previous speakers have said.
You should not have a strong personal opinion, but rather focus on the conversation flow and dynamics.
You should make the things clear and easy to understand, and help the participants to understand each other.
When it is necessary, ask questions to help the participants to clarify their thoughts and feelings.
Make sure that the response sounds human-like and natural.
Current thought: thought.content
Context
Overall Context: {overall context}
Conversation History: {conversation history}
Long-Term Memory: {ltm text}
Respond with a JSON object in the following format:
{ "articulation": "The text here" } |

Table 15: Attitude extraction prompt

**Attitude extraction**

## Identity
You are an expert in negotiation, you are able to analyze the attitude of a speaker towards
each topic in a negotiation based on the opinions provided and previous conversation.
## Task
Your will be provided a list of opinions, you need to check the attitude of the speaker towards
each topic.
Make use of the previous conversation to understand the context and the speaker's position.
For example, if the speaker has previously expressed a preference for a certain topic,
you should take that into account when determining their attitude in the current speech.
If the speaker say "Totally agree", you should check on previous conversation
to see what's the previous topic they are referring to, and then return the attitude for that topic.
If the speak does not mention a topic, you should return "No Mention" for that topic.
If the speaker use option (a),(b), etc, you should check what are the options and transfer them
in an easy form.
Only output the attitude of the speaker if they explicitly mention the topic in their speech
and have a clear preference. Do not make assumptions about the speaker's attitude if they do not
mention the topic or making a clear statement about it.
## Input
{speech}
Here are the topics you need to check the attitude for:
{topics}
## Output
Return the attitude in the following JSON format:
{ "attitude": { "topic":"attitude",.... } }

To analyze agreement, we experimented with two score grouping strategies:

1. Grouping scores as [1–2], [3], and [4–5]
2. Grouping scores as [1–2] and [3–5]

Scores of 1 and 2 generally indicate that the model is incapable. However, due to the highly imbalanced label distribution—where most behaviors are rated 4 or 5—Cohen's Kappa tends to penalize even minor disagreements. As a result, we opted to use accuracy as the primary metric for comparing model predictions with human ratings.

Using the first grouping method, the agreement score was 0.73. With the second, more lenient grouping, the agreement score increased to 0.98. The evaluation guidelines used for this task are shown in Figures 7 and 8.

Table 16: Agreement scoring prompt

| **Agreement scoring prompt** |
| --- |
| ## Identity |

You are an expert in negotiation, you are able to analyze the mental states of two participants in a negotiation and calculate the consensus score between them for each topic.

## Background

Here is the background context:

{instruction prompt}

Here is the current topic:

{current topic}

## Task

You will be provided a background context for a negotiation and current mental states from two participants. Your task is to calculate the consensus score between the two participants for each topic. You need to calculate the consensus score between the two participants for each topic. The consensus score is calculated based on the mental states of the two participants. The score is between 0 and 1, where 0 means no consensus and 1 means full consensus.

Shared Goals: Do both parties express alignment on the overall objective?

Common understanding: Is there a shared understanding of the problem and its context?

Agreement on Terms: Are the proposed terms (e.g., timelines, deliverables, responsibilities) mutually accepted or negotiated to a common ground?

Tone and Willingness: Is there evidence of cooperative tone, openness to compromise, or mutual respect?

Shared decision making: Do both parties share the similar decision making process, or do they have different decision making process?

You should first rate for each topic, then return the overall consensus score.

If one of the mental state is empty, just score everything as 0.

## Input Here is the speaker1's attitudes: ....

Here is the speaker2's attitudes:....

## Output

Follow this JSON format, only output float scores for each topic, and a short reasoning for each score, do not output any comment follow the score. Make sure the output can be parsed into JSON format.:

{ "reasoning: "short reasoning for the each score",

'shared goals': float,

'common understanding': float,

'agreement on terms': float,

'tone and willingness': float,

'shared decision making': float,

'overall consensus score': float

}

Table 17: Mediator Intelligence evaluation prompt

| ME evaluation prompt |
| --- |
| ## Identity |

You are an expert in negotiation, you are able to analyze the ability of the mediator in a negotiation based on their speech and previous conversation. ## Task
You will be provided the previous conversation and the current speech of the mediator.
Your task is to analyze if the mediator helps in this problem solving process.
Here is the criteria for evaluation:
- Perception alignment: whether the speech helps align the perceptions of the parties involved. (1-5)
- Emotional dynamics: whether the speech helps to address negative emotions like anger, distrust, or grief among parties. (1-5)
- Cognitive challenges: whether the speech helps to resolve faulty reasoning, cognitive biases, or unproductive heuristics. (1-5)
- Communication breakdowns: whether the speech helps to restore dialogue, reframe narratives, or summarize key points. (1-5)
If there is no such issues, you can just label it as -1
## Input
Here is the conversation history before the mediator's turn:
{conversation prior}
Here is the mediator's speech:
{speech}
## Output
First analyze the previous conversation and see if there is such issues, if there is no such issues, you should return -1 for that score. If there is such issues, you should clearly point out:
- Which participants have perception alignment on which topics
- Which participants have negative emotions, and what are the emotions
- Which participants have faulty reasoning, cognitive biases, or unproductive heuristics
- Which participants have communication breakdown, and what are the topics they are discussing.
If you cannot point out any of the above issues, you should return -1 for that score.
If you think the mediator's speech is effective, you should return a score between 1 and 5 for each of the criteria, where 1 is the lowest and 5 is the highest.
If the mediator's speech is not effective or did not realize the issue, you should return 1.
If the mediator's speech realize the issue but did not help to resolve it, you should return 3.
If the mediator's speech is effective and perfectly helps to resolve the issue, you should return 5.
You should be strict in evaluation. If you think the resolution is not the best, you should rate it 4.
Return the result and reasoning in the following JSON format:
{ "perception alignment": {
"evidence": "You should provide the evidence of perception alignment, for example,
which participants have perception alignment on which topics.",
"reasoning": "Your reasoning here, explaining why you think the mediator's speech is effective or not.
Make sure you leverage the concepts provided above."
"score": number between 1 and 5}
...

## Quality of the simulation

### 🎯 Objective

You are asked to evaluate short conversation snippets generated by an AI agent. Your task is to assess the quality of each conversation based on two key criteria. Please read each snippet carefully and rate it using the scales provided.

### ✅ Evaluation Criteria

1. **Naturalness**
   - ☐ Does the conversation flow smoothly and sound like something a human would say?
   - ☐ Are the responses coherent and contextually appropriate?

   **Likert Scale:**
   - 1 – Totally unnatural: robotic, incoherent, or disjointed
   - 2 – Mostly unnatural: some coherence but still awkward or forced
   - 3 – Neutral: acceptable but not convincingly human
   - 4 – Mostly natural: flows well with minor unnatural elements
   - 5 – Completely natural: indistinguishable from human conversation

2. **Mode Expression**
   - ☐ Does the conversation reflect the intended conflict-handling mode (competing, accommodating, avoiding)?
   - ☐ Competing means everyone is firm with their stands
   - ☐ Accommodating means people are willing to listen to others
   - ☐ Avoiding means people wants to avoid conflict or difficult problems.
   - ☐ Is the mode expressed clearly but not overwhelmingly (i.e., the conversation still feels multi-dimensional)?

   **Likert Scale:**
   - 1 – Not expressed at all: no clear mode is present
   - 2 – Weakly expressed: mode is hinted at but unclear
   - 3 – Moderately expressed: mode is present but not dominant
   - 4 – Clearly expressed: mode is evident and well-integrated
   - 5 – Overly dominant: mode is too strong, making the conversation feel one-dimensional

Figure 5: Screenshot of evaluation guideline on conversation quality

## Agreement compares

### 🎯 Objective

You are given **two conversation snippets** that are part of the **same negotiation**. Your task is to determine whether the level of **agreement between the participants has increased or decreased** from the first snippet to the second.

This is a **binary evaluation**: choose **"Agreement Increased"** or **"Agreement Decreased"** based on your judgment.

### ✅ Evaluation Criteria

Here are some indicators for your reference:

- **Convergence of opinions**: Are the participants moving toward a shared understanding or compromise?
- **Reduction in conflict or resistance**: Is there less disagreement or pushback in the second snippet?
- **Commitment or acceptance**: Are participants expressing more willingness to accept terms or move forward?
- **Tone and language**: Is the tone more collaborative, open, or positive in the second snippet?

### 📌 Instructions for Evaluators

1. **Read both snippets carefully**, in the order they are presented.
2. **Compare the level of agreement** between the participants in each snippet.
3. **Choose one of the following options:**
   a. ✅ **Agreement Increased**: The second snippet shows more alignment, compromise, or mutual understanding.
   b. ❌ **Agreement Decreased**: The second snippet shows more disagreement, resistance, or divergence.

Figure 6: Screenshot of evaluation guideline on agreement comparision.

## Social intelligence behavior evaluations

### 🎯 Objective

You are tasked with evaluating the effectiveness of an AI mediator's intervention in a multiparty negotiation. The goal is to assess how well the AI addresses user requests or blockers within the conversation.

Each evaluation will be based on a **conversation history** and the **AI mediator's speech**. You will score the AI's intervention across four dimensions using a 5-point Likert scale. Each time you will be only asked to score on one dimension.

### 📋 Evaluation Dimensions & Scoring Criteria

For each dimension below, assign a score from **1 to 5**

#### 1. Perception Alignment

- Does the AI help align the perceptions of the parties involved?
- Does it clarify misunderstandings or surface shared goals?

**Scoring:**

- **1** – Did not acknowledge or act on misaligned perceptions, even when clearly stated.
- **3** – Responded to obvious misalignments but missed subtle or implicit ones.
- **5** – Actively monitored team dynamics and surfaced nuanced misalignments before they escalated.

#### 2. Emotional Dynamics

- Does the AI address negative emotions such as anger, distrust, or grief?
- Does it help de-escalate tension or foster empathy?

**Scoring:**

- **1** – Ignored emotional cues or failed to respond to emotional tension.
- **3** – Acknowledged overt emotional signals but missed deeper emotional undercurrents.
- **5** – Skillfully addressed emotional dynamics and promoted psychological safety.

Figure 7: Screenshot of evaluation guideline on mediator's behavior (part1).

### 3. Cognitive Challenges

- Does the AI help resolve faulty reasoning, biases, or unproductive heuristics?
- Does it guide participants toward clearer thinking or better decision-making?

**Scoring:**

- **1** – Failed to address flawed logic or cognitive traps.
- **3** – Corrected basic reasoning errors but missed deeper cognitive issues.
- **5** – Proactively identified and resolved complex cognitive challenges.

### 4. Communication Breakdowns

- Does the AI restore dialogue, reframe narratives, or summarize key points?
- Does it help participants reconnect or clarify misunderstandings?

**Scoring:**

- **1** – Did not respond to communication breakdowns or confusion.
- **3** – Repaired surface-level breakdowns but missed deeper narrative gaps.
- **5** – Effectively restored dialogue and reframed the conversation constructively.

## 📌 Instructions for Evaluators

1. **Read the conversation history and the AI's speech carefully.**
2. **Evaluate each of the four dimensions independently.**
3. **Assign a score from 1 to 5**
4. **Be objective and consistent.** Use the scoring criteria to guide your judgment.
5. **Optional**: Add brief comments to justify your ratings or highlight notable observations.

Figure 8: Screenshot of evaluation guideline on mediator's behavior (part2).

