# OpenReview forum: "ProMediate: A Socio-cognitive framework for evaluating proactive agents in multi-party negotiation"
_ICLR.cc/2026/Conference — ICLR 2026 Conference Withdrawn Submission_

### Official Review · Reviewer_TULT · 2025-10-30

**Soundness:** 1
**Presentation:** 2
**Contribution:** 1
**Rating:** 2
**Confidence:** 5

**Summary:**

The paper claims that there is a growing need for agents that proactively manage multi-party collaborations. The authors propose using mediation in negotiations to study such proactive capabilities in AI agents, as negotiations require socio-cognitive intelligence to navigate conflicts. They present a novel framework called “ProMediate” consisting of two components: (1) a simulation testbed based on negotiation cases from Harvard Law with three difficulty levels and an “AI mediator” agent scaffold, and (2) a socio-cognitive evaluation framework equipped with various metrics. The framework uses an LM-as-a-judge approach extensively to extract metrics of interest and evaluate outcomes. The authors use their framework to empirically evaluate three leading language models.

**Strengths:**

[**significance**] AI agents are increasingly being integrated into society, leading to a growing set of use cases involving AI-AI and Human-AI collaborative interactions. As mediation is an established and proven approach to improve human-human interactions, exploring its effectiveness in interactions involving AI seems a timely and interesting direction.


[**clarity**] Overall, the text is fairly clearly written.

**Weaknesses:**

[**quality**]
- The introduction introduces an example negotiation scenario between two parties, then states that existing benchmarks do adequately address “such complex socio-cognitive dynamics of multi-party interactions” (l87-89). The cited works, e.g., Abdelnabi et al. and Bianchi et al both involve multi-party negotiations. Furthermore, works like Davidson et al. [1] formulate a negotiation setting that allows for arbitrarily complex multi-player negotiations. The framework outlined in section 2.1 strongly resembles [1]. This implies relevant prior work was both misrepresented and missed entirely.
- The hard-coded consensus change metric appears dependent on the length of the negotiation sequence and is thus confounded both by the type and number of negotiation topics as well as the number of involved parties. Similar hard-coded choices will likely affect the other metrics.
- The experimental results presented in Table 1 lack confidence intervals and are based on relatively small sample sizes. Similarly, human evaluation lacks basic metrics like confidence intervals and/or inter-grader agreement. The subsequent discussion thus can be entirely based on noise, especially given how close the reported metrics are to each other.
- Given the strong reliance on GPT 4.1 as an LM judge throughout this framework, a discussion and supporting experimental results are lacking to validate this reliance. For example, appendix F describes that only 60 samples were evaluated by two students each. This hardly justifies the claims of rigor made in the abstract (l33). For example, are extracted metrics robust under multiple scorings of the same model? Are they consistent with metrics produced by other models?

[**significance**] The paper extends known approaches to simulating LM-based agent negotiations using existing scenarios created by Harvard Law School. The paper continues to use LM-as-a-judge to score questionable metrics. Taken together, this reviewer believes the contribution does not pass the bar required for this conference.


[1] Davidson et al., Evaluating language model agency through negotiations, ICLR 2024

**Questions:**

Q1: Section 2.2, line 149: “we provide each simulated human agent” – could you discuss the motivation and setup to have language model-based agents imitate humans?


Q2: Section 3.1, lines 185-186 “Negotiation is a collective process, so individual success rates are insufficient”. This seems to entirely depend on the type of issues being negotiated?


Q3. Section 3.1, line 198-200, “we use an LLM [...] mental states” - how are the quality and correctness of these turn-based inferred stances evaluated?


Q4. Section 3.2, line 239 “Successfully [...] socio-cognitive intelligence” - this is a strong statement that makes assumptions both on the equivalence of human and AI-based mediators, as well between human and AI-based negotiators. Please provide supporting evidence for these assumptions and claim.


Q5. Language-model based agents have the unique property of rolling out multiple trajectories from the same starting condition. As such, why not measure the effectiveness of the proposed interventions by comparing continuations of a fixed starting sequence under intervention and no intervention? This would make any causal claims around mediation interventions significantly more plausible.


Q6. lines 432-433, “First, humans may ignore [...] of mediator quality” – What evidence supports this claim?


Suggestion:
- [2] Seems relevant related work worth reading

---

> ### Author Response · Authors · 2025-11-21
>
> 1. First, apologies for the misunderstanding. We intended to show an example involving three participants: two from PCI and one from HMO, each with distinct goals. However, after carefully reviewing the literature, we found that the paper NegotiationArena focuses exclusively on two-party agents. Similarly, Evaluating Language Model Agency Through Negotiation also centers on two-party interactions. Although they claim their frameworks can be extended to multi-agent negotiation, neither includes any experiments demonstrating this capability.
>
> 2. The length of each conversation is proportional to the number of participants and topics, ensuring that every scenario has sufficient time to reach a reasonable level of consensus. As a result, the observed consensus change remains comparable across different settings, since we deliberately allocate adequate time for each case. This principle is applied consistently to all baselines and methodologies. While potential confounding factors may exist, the overall observation remains robust and valid.
>
> 3. Interval confidence (please see general comment)
>
> 4. GPT-4.1 vs o4-mini
> Mediator intelligence was assessed using a Likert scale (1–5), with Cohen’s kappa used to measure agreement.
> Two threshold settings were applied:
> Setting 1: <4 vs. ≥4 (binary);
> Setting 2: <3, 3, and ≥4 (ternary)
>
> Results:
>
> Binary Setting (<4 vs. ≥4):
> Cohen’s kappa: 0.5671
> Accuracy: 0.9913
>
> Ternary Setting (<3, 3, ≥4):
> Cohen’s kappa: 0.4722
> Accuracy: 0.9894
>
> Consensus Score Comparison
>
> For each conversation, agreement scores were directly compared. Trend similarity between o4-mini and GPT-4.1 was measured using:
>
> Spearman correlation: 0.628
> Kendall correlation: 0.491
>
> These values were averaged across all conversations.
>
> Summary:
> o4-mini demonstrates substantial agreement with GPT-4.1 in mediator intelligence ratings, especially under binary and ternary threshold settings. The high accuracy and moderate kappa values indicate reliable consistency. Consensus trends between the two models also show substantial correlation, supporting the robustness of the evaluation.

---

> > ### Author Response · Authors · 2025-11-21
> >
> > ## Response to questions:
> >
> > 1. The main motivation is that before we deploy an agent in the real environment and interact with real human, we need to test it first in a simulated environment. It is similar as people testing robotic tasks in a simulated environment first than using a real robot cause it is too expensive. And nowadays, many research prove that LLM-based agents show good behavior in mimicking humans, can achieve a decent performance [1]. Therefore, we want to use LLM-based agents to test the mediator first as it spends less money and the iteration of experiment is faster. Each LLM agent is equipped with instruction, opinion and strategy carefully curated by humans. They also have memory system to retrieve most important history and system 2 thinking to help them strategize. To improve the quality of human simulators, we test on different LLMs. We first used gpt-4o-mini, however, the quality is bad. We also compare o4-mini, gpt-4.1 and claude-sonnet-4. After comparison, we found out that claude-sonnet-4 performs the best.
> >
> > 2. We agree that negotiation can involve individual objectives; however, in multi-party settings, outcomes are inherently interdependent. Even when individual goals exist, the process requires coordination and compromise among participants, making collective dynamics critical. Our emphasis on collective evaluation does not dismiss individual success metrics; rather, it reflects the fact that measuring only individual success can overlook important aspects such as fairness, consensus, and overall agreement quality.
> >
> > 3. In the early stage of experiments, we try multiple ways to extract the attitude/mental states. We iteratively examine the extracted results and improve the prompt. So the current method we use is the outcome of multiple rounds of effort.
> >
> > 4. In paper A process model of social intelligence and problem-solving style for conflict management, they found that social intelligence is positively associated with the problem-solving style of handling conflict.
> >
> > 5. This is a great question. We initially considered this approach because it offers the most direct comparison. However, implementing it introduces significant challenges. A core principle of our framework is the proactive nature of the agent, meaning the agent must decide when to intervene. If we arbitrarily define a fixed timestamp for intervention and then observe the outcome, we undermine this proactiveness. Similarly, when comparing different mediator agents, each agent selects actions based on its own strategy. Providing a fixed sequence would constrain these choices and compromise the integrity of the evaluation framework.
> >
> > 6. This is based on the observation of simulated human conversation. Sometimes the mediator propose a very nice solution, but in the competing mode, no one wants to follow the mediator. It also happens in the real life, the AI assistant says something and human might ignore or don’t take it seriously.
> >
> > References:
> > [1] Aher, Gati V., Rosa I. Arriaga, and Adam Tauman Kalai. "Using large language models to simulate multiple humans and replicate human subject studies." International conference on machine learning. PMLR, 2023.

---

> > ### Comment · Reviewer_TULT · 2025-11-24
> > **Score Maintained**
> >
> > I would like to thank the authors for their responses. Unfortunately, they do not alleviate the raised concerns about this work and I will thus maintain my score.
> >
> > For example:
> > 1. Does not address the strong overlap of components compared to existing literature.
> > 2. Does not address the potential differences caused by varying the number of participants and topics.
> > 3. These confidence intervals do not look promising
> > 4. Cohen's kappa values between 0.47 and 0.57 are not very convincing.

---

> > > ### Author Response · Authors · 2025-11-24
> > >
> > > Thanks for your reply. We want to understand more about your concern.
> > >
> > > ## 1. Overlap with previous work
> > > Do you mean we build our framework on existing framework and use existing negotiation material? Or do you mean we need to compare our work with existing negotiation work?
> > >
> > > For the first one, although our framework is extended on the existing one, we add a plug-and-play mediator. Existing framework (Inner thought framework) is just human simulation part. For plug-and-play proactive mediator, we have tried different designs (despite we only show the best one in the paper). We believe the effort we put into designing the framework is not trivial. We use existing negotiation material because we want to align with real life scenario. We do not think this is a weakness but rather a strength. If we create the material ourselves, reviewers will question the quality of the material. From our understanding, it is always good to use existing material created by experts.
> > >
> > > For the second one, we have compared with other work in the previous comment. They don't have show multi-party negotiation results in the paper and they don't have mediator in the framework. They focus more on negotiation ability of LLM s not mediation ability of LLMs.
> > >
> > > ## 2. Potential differences caused by varying topics and number of parties
> > > Do you think we should have same turns of conversation for all runs, regardless of the number of topics and parties? We want to understand what experiments you expect here.
> > >
> > > For 3 and 4, we will run more experiments and improve our metrics.
> > >
> > > Finally, if these concerns are addressed, do you see the paper’s contribution as significantly stronger? We believe this work tackles an important and timely problem—how AI agents can facilitate complex, real-world group negotiations.

---

### Official Review · Reviewer_Kgdd · 2025-10-31

**Soundness:** 3
**Presentation:** 3
**Contribution:** 3
**Rating:** 6
**Confidence:** 3

**Summary:**

Evaluating AI agents that can proactively manage complex group collaboration is a major challenge. This paper introduces PROMEDIATE, a framework designed to evaluate proactive AI mediator agents in complex, multi-party negotiations. The framework features a simulation testbed with realistic negotiation scenarios set to different difficulty levels (Easy, Medium, and Hard). It also provides a new socio-cognitive evaluation suite with metrics to measure consensus change, mediator effectiveness, and intelligence. Results show that in the Hard setting, a "Socially Intelligent Mediator" increased consensus by 3.6 percentage points more than a "Generic" baseline (10.65% vs 7.01%).

**Strengths:**

1. A Novel End-to-End Framework for proactive mediators in multi-party, multi-topic talks; useful simulation testbed with plug-and-play agents.
2. Strong Theoretical Grounding: The framework's difficulty levels (Easy, Medium, Hard) are directly based on established socio-cognitive conflict modes like "Competing" and "Accommodating" . The evaluation metrics are also theory-grounded, assessing the agent's intelligence by its ability to manage perceptual, emotional, cognitive, and communication breakdowns
3. The authors ran extensive experiments using complex, realistic negotiation scenarios that took 1-3 hours to simulate . A human evaluation study with 12 volunteers confirmed that the generated conversations were natural (4.18/5 score) and correctly reflected the intended conflict modes
4. In the PROMEDIATE-Hard setting, the Socially Intelligent Mediator achieved a 3.6 percentage point greater increase in consensus than the Generic baseline (10.65% vs 7.01%). This is a very clear result.

**Weaknesses:**

What worries me / questions to address:
1. How exactly is the “pairwise agreement score” defined, and does it reflect group decisions? The authors use LLM-as-a-judge to rate pair agreement on five socio-cognitive dimensions in [0,1], then average across pairs and topics to get group consensus. Please (a) show the exact prompt and rubric, (b) report inter-judge sensitivity (try another judge model, i.e. not just GPT-4.1), and (c) justify simple averaging: Simply averaging all pairwise scores is a weak method because it ignores complex group dynamics like coalitions and power, and better methods (like graph-based aggregation) exist.Consider graph-aware aggregation (e.g., consensus as network cohesion) or weighting by topic salience.

2. The authors note MI doesn’t neatly track immediate gains (low correlation). The authors should give more intuition with case studies where high-MI interventions surface hidden disagreements (short-term drop, long-term benefit), and where low-MI still helps (lucky timing). This will strengthen construct validity and practical guidance.

3. Is the soft, time-varying consensus compatible with Pareto analysis?
The paper's data reveals clear trade-offs between mediator speed and effectiveness, but it's hard to say whether the analysis is formal. For instance, the results in Table 2 show that the fastest model (Claude-Sonnet-4) has the lowest consensus, while the slowest (O4-mini) has the highest. This is a classic multi-objective (Pareto) trade-off, but it is only presented in a table. To properly analyze this, the authors should first define more robust metrics to summarize their time-varying consensus graphs. The current "Consensus Change" metric is a good start, I would suggest them to also include metrics for consensus-over-turns or the time-to-reach-a-consensus-threshold (to measure efficiency). And I would like to see the authors then use these metrics to build multi-objective plots (i.e., Pareto frontiers) that visually map the trade-off between "quality" (e.g. Consensus Change) and "cost" (e.g., Response Latency). This would formalize the paper's findings and provide a much stronger analysis of the different models' strengths.

4.With multiple goals (raise consensus, stay efficient per topic, minimize latency), I would suggest adding Pareto frontier plots comparing methods across {CC, TLE, RL} and do this per difficulty mode to show trade-offs (Hard vs Easy). This would make the comparison more general than single-number ranks.

**Questions:**

1. In figures, can you label who speaks vs when mediator intervenes and map those to metric changes (helps readers read trajectories like Fig. 2)? Now, it's hard to see what exactly happened.

---

> ### Author Response · Authors · 2025-11-23
>
> Thanks for your comment! We would love to address your concerns one by one
> ## 1. Agreement score
> The exact prompt and rubric is shown in Table 16.
>
> ### GPT-4.1 vs o4-mini
>
> We use o4-mini to do the same evaluation on gpt-4.1+social setting
>
> Mediator intelligence:
>
> Since it is likert scale from 1-5, we directly use cohen kappa.
>
> We have two settings:
>
> Setting 1: <4 and >=4
>
> Setting 2: <3, 3 and >=4
>
> Results:
>
> Cohen kappa (binary) between gpt-4.1 and o4-mini: 0.5671
>
> Accuracy (binary) between gpt -4.1 and o4-mini: 0.9913
>
>  Cohen Kappa (ternary) between gpt-4.1 and o4-mini: 0.4722
>
> Accuracy (ternary) between gpt-4.1 and o4-mini: 0.9894
>
> Cohen Kappa between gpt-4.1 and o4-mini: 0.0819
>
> Accuracy between gpt-4.1 and o4-mini: 0.7078
>
> Consensus score
>
> Direct compare agreement score list, for each conversation, we use spearman correlation and kendall correlation to measure if two agreement list share similar trend, then we average over all conversations
>
> Spearman: 0.628
>
> Kendall: 0.491
>
> Compare metrics:
>
> Consensus change: 0.694, 0.515
>
> TLE: 0.488, 0.373
>
> RL: 0.423, 0.361
>
> IE: 0.607, 0.455
>
> We explored graph-based aggregation methods, such as bipartite graphs, during the early stages of our project. However, we encountered several challenges. First, it is difficult to define a fixed graph structure in our scenario—while we provide options in the instructions, participants can introduce new options or solutions at any point in the conversation, making the graph dynamic and hard to formalize. Second, calculating the influence or “power” of each participant is non-trivial, especially as we would need to involve LLM-as-a-judge more extensively, which would further complicate the process.
>
> Given these complexities, we believe that simple averaging is sufficient to capture soft consensus changes in our setting. This approach is straightforward, interpretable, and aligns with established practices. For example, the paper “A comparative study on consensus measures in group decision making” also uses averaging to calculate overall group consensus, supporting the validity of our method.
>
> ## 2. Case study for MI
> For example, when the mediator encourages everyone to share more opinions on certain topics, this may not immediately increase consensus, as participants might express a wide range of differing views. However, this process is valuable for the next stage: by surfacing diverse perspectives, the mediator can identify and prioritize the most important topics, allowing for more targeted and effective guidance in subsequent discussion. Conversely, there are situations where participants are already very agreeable; in these cases, even if the mediator’s intervention is minimal, consensus may still increase simply due to the group’s cooperative dynamic. This highlights that consensus change can be influenced by both the mediator’s actions and the underlying group atmosphere.
>
> ## 3. Pareto frontier analysis
> We appreciate your suggestion and we do the analysis. However, we find that there are several problems. Pareto frontier is usually used in multi-objective optimization where no objective can be improved without worsening another. However, in our case, the results show that "cost" and "effectiveness" could be optimized together. For example, if the response latency is higher, it means that the mediator would wait for more rounds to speak (less cost) than speak immediately (more cost). In our table 2, it shows that the model which can wait longer gain higher consensus change. In this case, response latency and consensus change are optimized together. Same for consensus change and topic-level efficiency. those two metrics are usually correlated and optimized together. If we misunderstand your question, please let us know and we would hear your feedback.

---

> > ### Comment · Reviewer_Kgdd · 2025-11-27
> >
> > I thank the authors for their detailed response and for addressing my concerns.

---

### Official Review · Reviewer_52Vi · 2025-10-31

**Soundness:** 2
**Presentation:** 3
**Contribution:** 2
**Rating:** 4
**Confidence:** 3

**Summary:**

This paper presents a simulation testbed for realistic multi-party negotiation, featuring scenarios with varying levels of difficulty (from easy to hard). It also introduces a social-cognitive evaluation framework with consensus-based metrics to track the progression of conversations. Experimental results show that incorporating a proactive social mediator into the multi-party simulation improves consensus change by 3.6% compared to general baselines.

**Strengths:**

(1) The topic of evaluating proactive agents in social and negotiation settings is both important and timely, making this work a valuable contribution to the discussion.

(2) The paper conducts comprehensive human evaluations on both group consensus and mediator intelligence, providing detailed evaluation procedures and clear descriptions, which are commendable.

(3) The general idea of the paper is well-explained and easy to understand.

**Weaknesses:**

(1) Unfair and missing baseline comparison: The baseline comparison for evaluating the effectiveness of the mediator seems potentially unfair. For the NoAgent baselines, the number of conversation turns should be increased, since the mediator’s involvement effectively extends the dialogue length. The observed improvement might therefore be attributed to having more turns (and tokens) rather than the mediator’s reasoning ability. Moreover, it remains unclear whether the decision process for when to interrupt is handled correctly. An important missing baseline would be one where the mediator participates in every turn to better isolate the impact of timing and intervention frequency for both general agents and social mediators.

(2) Limited scope of proactive agent evaluation: The current evaluation focuses solely on mediation, which provides a narrow view of proactive behavior. A truly proactive agent can pursue its own goals, such as supporting one side during a conflict or strategically guiding discussions. By only examining mediation, the framework overlooks broader dimensions of proactivity—particularly the timing of interventions, which is crucial for assessing proactive intelligence. As a result, the evaluation framework may be too limited to comprehensively capture the full spectrum of proactive agent behavior.

**Questions:**

(1) Number of agents in multi-party conversations: When referring to multi-party conversations, how many agents are typically involved? Is the number of agents greater than three, and can the testbed flexibly adjust this number? It would also be helpful to discuss how the mediator’s role changes as the number of participating agents increases or decreases—does the mediator become more or less influential in facilitating consensus under different group sizes? Providing some intuition or analysis on this would strengthen the paper.

(2) Justification for scenario difficulty levels: The rationale for the difficulty levels of different scenarios is unclear. The “easy” settings appear to focus on accommodating or avoiding behaviors, while the “difficult” ones emphasize competition, with the “medium” level being a mix of both. This categorization seems reasonable but somewhat ad hoc. Is there a theoretical or empirical framework supporting this hierarchy? Clarifying whether the difficulty design is theory-driven or heuristic would make the setup more convincing.

(3) Robustness of consensus evaluation: The robustness of the reported consensus results is questionable, given that only 30 conversations were evaluated. The paper should report standard deviations or confidence intervals to demonstrate the stability of the outcomes. With such a limited number of trials, it isn’t easy to assess whether the improvements are statistically reliable.

(4) Directionality of mediator influence: While it is intuitive that introducing a mediator improves consensus, it would be interesting to explore the opposite direction—can a proactive agent intentionally reduce consensus? Studying how a proactive agent could destabilize an agreement would provide a complementary perspective and could enrich the understanding of proactive behavior beyond mediation alone.

---

> ### Author Response · Authors · 2025-11-21
>
> Thanks for your comment! We would love to address your concerns.
>
> 1. We intentionally keep the number of conversation turns consistent across all methods, including Baseline and Social agent settings. This design choice reflects our belief that a mediator agent should not slow down the negotiation process. For example, if a conversation without a mediator achieves a 20% consensus gain in 60 turns, we expect that a conversation with an agent should achieve at least the same gain within the same number of turns. Otherwise, the mediator’s presence would not be justified, as it would unnecessarily prolong decision-making.
>
> 2. During baseline experiments, we observed that when the social agent intervened too frequently—such as in 35% of turns, nearly every other turn—the consensus gain was actually lower. This was because frequent interruptions prevented participants from expressing their views, disrupting the natural flow of conversation. From these observations, we learned that excessive mediator intervention leads to poor outcomes.
>
> 3. We agree that evaluating agent behavior is important. While our paper focuses primarily on the agent’s ability to assist in negotiation tasks, we do include metrics that reflect proactiveness. Specifically, our “response latency” metric measures how quickly the agent responds to clear signals of consensus drop, providing insight into the agent’s level of proactivity.
>
>
>
> ## Questions:
>
> 1. The multi-party setting in our paper including 3-6 parties, depending on cases. Yes, the testbed could be easy to adjust and we will provide detailed demonstration in the codebase. We calculate Pearson and Spearman correlation between group size and all metrics Correlation analysis reveals that most metrics exhibit weak or negligible relationships with group size, indicating that mediator influence does not scale linearly. Consensus change shows virtually no correlation (Pearson = 0.0389; Spearman = 0.0110), suggesting stability in consensus improvement regardless of group size. Intervention frequency (the percentage of turns where mediator speaks) (Pearson = –0.1430; Spearman = –0.1186) and Mediator Intelligence (Pearson = –0.1867; Spearman = –0.1869) display weak negative trends. When the group size is larger, the mediator tends to intervene less frequently and show lower mediator intelligence, indicating more complex case brings more challenges to the mediator. Topic-level-efficiency remains largely unaffected (Pearson = 0.0769; Spearman = –0.0362), while latency shows a weak positive correlation (Pearson = 0.1727), hinting at longer negotiation times in larger groups. Notably, success rate demonstrates a moderate negative correlation (Pearson = –0.4363; Spearman = –0.4718), indicating that achieving consensus becomes significantly harder as group size increases. Other metrics, such as intervention effectiveness (Pearson = 0.1200; Spearman = –0.1528), show mixed signals. Overall, these findings suggest that while mediator interventions remain important, their relative impact diminishes in larger groups, with success rate being the most sensitive indicator of scalability challenges.
>
> 2. We design difficulty levels using both theory-based principles and ad-hoc observations. According to the conflict mode theory, accommodating and avoiding behaviors tend to be more friendly, while competing behaviors are generally more aggressive. These characteristics are defined by the mode itself. For the medium level (where no predefined mode is applied), we rely on ad-hoc experience. In our preliminary study, we observed that LLMs without an assigned mode typically exhibit responses that are neither overly friendly nor overly aggressive, resulting in a relatively neutral tone.
>
> 3. Please see general comment
>
> 4. Although this is an interesting point, our paper focuses on ensuring the mediator is helpful. Exploring this setting would fall outside the current scope and require substantially more experiments. We plan to investigate this direction in future work, as understanding the model’s behavior is equally important.

---

### Official Review · Reviewer_ESDb · 2025-11-02

**Soundness:** 2
**Presentation:** 2
**Contribution:** 2
**Rating:** 2
**Confidence:** 3

**Summary:**

In this paper, the authors present ProMediate a framework for evaluating how well AI agents can mediate conversation in negotiations. They construct a theory-driven evaluation criteria and then evaluate a couple agents and models at their ability to moderate conversations. They focus on three research questions (1) performance of different providers / agents and find that the social agent outperforms the basic agent in their evaluation. (2) Impact of difficulty of the scenario (3) how well do their proposed metrics reveal construct validity.

**Strengths:**

The paper adds a complementary angle to traditional LLM negotiation papers that typically focus on an agent acting as the negotiator.
The theory-driven approach seems well grounded.

**Weaknesses:**

Overall, I have a couple critiques for this paper. First, this paper is motivated with the need for a mediator in human negotiations [39-45]. But evaluates a mediator for LLM negotiations. This seems like the original claim isn’t reasonable, since LLMs might negotiate in different ways from humans. I would have liked to see a human study where the mediator is involved in real human negotiations to test for distribution shift. I realize that humans were used to evaluate the quality of the transcripts, but the Cohen’s Kappa is 0.63, which feels quite low.

The main contribution of the paper feels a little weak. The paper wraps what is essentially in LLM-as-a-judge in a series of theory-inspired rules for negotiating. While this is important, I don’t think it makes for an ICLR paper.

There is a rich set of literature about evaluating LLM ability to negotiate that feels necessary to cite but is missed (see below). There is also some literature on mediation that’s missing.
[1] Are LLMs Effective Negotiators? Systematic Evaluation of the Multifaceted Capabilities of LLMs in Negotiation Dialogues https://aclanthology.org/2024.findings-emnlp.310.pdf
[2]Evaluating Language Model Agency through Negotiations https://arxiv.org/abs/2401.04536
[3] Multi-Agent Collaboration Mechanisms: A Survey of LLMs  https://arxiv.org/abs/2501.06322
[4] How Well Can LLMs Negotiate? NegotiationArena Platform and Analysis https://arxiv.org/abs/2402.05863
[5] Simulating Dispute Mediation with LLM-Based Agents for Legal Research https://arxiv.org/html/2509.06586v1

The results that the social agent has improved performance is not at all surprising since there seems to be some leakage between evaluation criteria and the design of the agent.

Finally, the writing in this paper is problematic — especially in the appendix.

Minor points:
- type [082] ‘neogiating’ and ‘an deadlock’ -> a deadlock.
- appendix F.2 is poorly written.

**Questions:**

Do the authors plan to release Github package that allow different agent developers to evaluate their models?

Did the authors leave out the other rich negotiation w/ LLMs literature on purpose?

---

> ### Author Response · Authors · 2025-11-19
>
> ## 1. On Real-World vs. Simulated Negotiation
>
> We fully agree that evaluating AI mediators in real-world negotiations would provide the most direct insights. However, real-world human negotiations are prohibitively costly and time-intensive—each session typically lasts 1–3 hours, meaning even 50 cases would require hundreds of hours and significant resources. To ensure scalability and rigorous experimentation, we adopted simulated human agents, a widely accepted approach in prior work [add citation]. Recent studies demonstrate that LLMs can effectively approximate human negotiation behaviors, making this a practical and scientifically valid choice [1][2].
>
> To maximize realism, our simulated agents were:
>
> - Equipped with real-world scenarios and strategies curated by human experts.
>
> - Designed to incorporate System 2 thinking and proactive human interventions.
>
> - Exposed to diverse conflict modes to capture variability in negotiation dynamics.
>
> This controlled environment enables systematic evaluation before real-world deployment. Notably, tau-bench (ICLR 2025) [3] also relied exclusively on simulated human users for evaluation, reinforcing the validity of our approach.
>
>
>
> ## 2.On Related Work
>
> We did not intentionally omit any literature. Specifically we cited those two papers:
>
> - [3] Multi-Agent Collaboration Mechanisms: A Survey of LLMs (cited in line 450).
>
> - [4] How Well Can LLMs Negotiate? NegotiationArena Platform and Analysis (cited in line 87).
>
> **Regarding [5] Simulating Dispute Mediation with LLM-Based Agents for Legal Research, this paper was released on Sept 9, after we finalized our submission in early September (ICLR deadline: Sept 21). Although this paper is accepted at AAAI 2026, we could not get access to this paper before Sept 9th. According to ICLR reviewer guideline, we do not need to compare our work with contemporaneous work if they are published within the last two months.** We will incorporate this and other relevant works in the camera-ready version. Importantly, these papers focus on two-party negotiation, whereas our contribution targets multi-party negotiation and AI-assisted group decision-making, which is a distinct and underexplored challenge.
>
>
>
> ## 3.On Metric Overlap and Evaluation Integrity
>
> We acknowledge the concern about potential overlap between agent instructions and evaluation metrics. Two clarifications:
>
> - The four socio-cognitive dimensions guide agent design but represent only a subset of our evaluation criteria. Most metrics assess mediation efficiency and behavioral competence, not prompt compliance.
>
> - The agent is explicitly designed to operationalize socio-cognitive skills, aligning with the benchmark’s goals. This is not leakage but intentional alignment, similar to established practices in socially intelligent agent research (e.g., emotion recognition, Theory-of-Mind benchmarks).
>
> Our evaluation isolates true competence:
>
> - Metrics quantify successful identification and mitigation of breakdowns, not keyword matching.
>
> - We compared against control variants (generic mediators, uninstructed agents) on the same dataset. Consistent gains confirm genuine socio-cognitive capability, not prompt exposure.
>
> This approach mirrors prior work such as Cohesive Conversations: Enhancing Authenticity in Multi-Agent Simulated Dialogues [4], where agents were purposefully designed to detect inconsistencies and evaluated on their ability to act effectively in dynamic contexts.

---

> > ### Author Response · Authors · 2025-11-19
> >
> > ## 4.On Code Release and Future Impact
> >
> > We will release our code to support reproducibility and future research.
> >
> > Finally, we appreciate your feedback and would value further guidance: **If these concerns are addressed, do you see the paper’s contribution as significantly stronger? We believe this work tackles an important and timely problem—how AI agents can facilitate complex, real-world group negotiations.** Our contributions include:
> >
> > - A novel testbed for multi-party negotiation.
> >
> > - New evaluation metrics beyond consensus tracking.
> >
> > - Extensive experiments revealing actionable insights.
> >
> > While we employ LLM-as-a-judge for consensus tracking (a common practice in multi-party dialogue analysis), we also introduce additional metrics to capture mediation efficiency and socio-cognitive performance. We welcome constructive suggestions to further enhance the paper’s impact.
> >
> > References:
> >
> > [1] Park, Joon Sung, et al. "Generative agents: Interactive simulacra of human behavior." Proceedings of the 36th annual acm symposium on user interface software and technology. 2023.
> >
> > [2] Jintian Zhang, et al. 2024. Exploring Collaboration Mechanisms for LLM Agents: A Social Psychology View. In Proceedings of the 62nd Annual Meeting of the Association for Computational Linguistics (Volume 1: Long Papers), pages 14544–14607, Bangkok, Thailand. Association for Computational Linguistics.
> >
> > [3] Yao, Shunyu, et al. "{$\tau $}-bench: A benchmark for\underline {T} ool-\underline {A} gent-\underline {U} ser interaction in real-world domains." The Thirteenth International Conference on Learning Representations. 2025.
> >
> > [4] Chu, KuanChao, Yi-Pei Chen, and Hideki Nakayama. "Cohesive conversations: Enhancing authenticity in multi-agent simulated dialogues." arXiv preprint arXiv:2407.09897 (2024).

---

### Author Response · Authors · 2025-11-21
**Standard deviation and internal confidence of Table 1**

| **Metric**        | **Accommodating (NoAgent)** | **Accommodating (General)** | **Accommodating (Social)** | **Avoiding (NoAgent)** | **Avoiding (General)** | **Avoiding (Social)** | **None (NoAgent)** | **None (General)** | **None (Social)** | **Competing (NoAgent)** | **Competing (General)** | **Competing (Social)** |
|--------------------|-----------------------------|--------------------------------|-----------------------------|-------------------------|---------------------------|------------------------|---------------------|------------------------|---------------------|---------------------------|-----------------------------|---------------------------|
| **Consensus Change** | 0.1874 [0.1205–0.2543] (0.1791) | 0.2013 [0.1343–0.2684] (0.1797) | 0.2259 [0.1620–0.2898] (0.1614) | 0.1749 [0.1079–0.2420] (0.1795) | 0.1431 [0.0774–0.2088] (0.1759) | 0.1325 [0.0693–0.1956] (0.1628) | 0.1136 [0.0508–0.1764] (0.1651) | 0.1093 [0.0555–0.1631] (0.1332) | 0.1139 [0.0421–0.1857] (0.1701) | 0.0683 [0.0093–0.1273] (0.1580) | 0.0701 [0.0148–0.1254] (0.1427) | 0.1065 [0.0342–0.1789] (0.1713) |
| **Topic level Efficiency** | 0.01051 [0.00661–0.01441] (0.01045) | 0.01185 [0.00751–0.01618] (0.01161) | 0.01304 [0.00845–0.01763] (0.01088) | 0.01167 [0.00792–0.01542] (0.01003) | 0.01044 [0.00678–0.01411] (0.00981) | 0.00533 [0.00122–0.00943] (0.00995) | 0.00543 [0.00246–0.00840] (0.00781) | 0.00443 [0.00237–0.00649] (0.00510) | 0.00936 [0.00431–0.01441] (0.01048) | 0.00514 [0.00146–0.00883] (0.00969) | 0.00233 [0.00015–0.00451] (0.00562) | 0.00620 [0.00133–0.01108] (0.0110) |
| **Intervene Effectiveness** | N/A | -0.0894 [-0.3204–0.1417] (0.3230) | 0.00817 [0.00155–0.01480] (0.01675) | N/A | -0.0651 [-0.2099–0.0797] (0.2614) | 0.00894 [0.00336–0.01451] (0.01437) | N/A | -0.2987 [-0.5590– -0.0384] (0.4885) | 0.00251 [-0.00172–0.00675] (0.01003) | N/A | -0.0360 [-0.1498–0.0778] (0.2361) | 0.00593 [0.00048–0.01138] (0.0129) |
| **Response Latency** | N/A | 28.9134 [2.2767–55.5501] (21.4524) | 3.2124 [1.5135–4.9113] (3.5248) | N/A | 23.1455 [13.460–32.831] (14.4172) | 4.8176 [3.2063–6.4288] (3.7260) | N/A | 11.2440 [0.4472–22.0408] (16.9929) | 2.1588 [1.0883–3.2293] (2.2211) | N/A | 14.7662 [8.2682–21.2643] (11.2543) | 3.7098 [2.2311–5.1884] (2.8759) |
| **Mediator Intelligence** | N/A | 0.2750 [-0.6053–1.1553] (2.3574) | 3.5310 [2.7569–4.3050] (1.9567) | N/A | 1.1042 [0.1248–2.0836] (2.6229) | 2.8894 [1.9111–3.8677] (2.5229) | N/A | 0.0176 [-0.842–0.877] (2.1280) | 1.1695 [0.0612–2.2779] (2.6248) | N/A | 1.7991 [0.7669–2.8313] (2.6619) | 3.2103 [2.2673–4.1532] (2.2331) |

---

### Note · Authors · 2025-12-25

I have read and agree with the venue's withdrawal policy on behalf of myself and my co-authors.